



# Demonstrating the impact of integrated drought policies on hydrological droughts

Doris E Wendt[1], John P Bloomfield[2,*], Anne F Van Loon[3,*], Margaret Garcia[4], Benedikt Heudorfer[5], Joshua Larsen[1], and David M Hannah[1]

[1]School of Geography, Earth and Environmental Sciences, University of Birmingham, Birmingham, UK
[2]British Geological Survey, Wallingford, UK
[3]Institute for Environmental Studies (IVM), Vrije Universiteit Amsterdam, NL
[4]Arizona State University, School of Sustainable Engineering and the Built Environment, Tempe, AZ, USA
[5]UDATA GmbH, Neustadt a.d. Weinstrasse, DE
[*]These authors contributed equally to this work.

**Correspondence:** Doris Wendt (DorisEWendt@gmail.com)

**Abstract.** Managing water-human systems in times of water shortage and droughts is key to avoid overexploitation of water resources, particularly for groundwater, which is a crucial water resource during droughts sustaining both environmental and anthropogenic water demand. Drought management is often guided by drought policies to avoid crisis management and to actively introduce management strategies during droughts. However, the impact of drought management strategies on hydro-
logical droughts is rarely assessed. In this study, we present a newly developed socio-hydrological model, simulating feedbacks between water availability and managed water use over three decades. Thereby, we aim to assess the impact of drought policies on both surface water and groundwater droughts. We tested this model in an idealised catchment based on climate data, water resource management practices, and drought policies in England. The model includes surface water storage (reservoir), groundwater storage for a range of hydrogeological conditions and optional imported surface water or groundwater. These
modelled water sources can all be used to satisfy anthropogenic and environmental water demand. We tested four aspects of drought management strategies: 1) increased water supply, 2) restricted water demand, 3) conjunctive water use, and 4) maintained environmental flow requirements by restricting groundwater abstractions. These four strategies were evaluated in separate and combined scenarios. Results show mitigated droughts for both streamflow and groundwater droughts in scenarios applying conjunctive use, particularly in low groundwater storage systems. In high groundwater storage systems, maintaining
environmental flows reduces hydrological droughts most. Scenarios increasing or restricting water demand have an opposing effect on droughts, although these scenarios are in balance when combined at the same time. Most combined scenarios reduce the severity and occurrence of hydrological droughts given an incremental dependency on imported water that satisfies up to a third of the total anthropogenic water demand. The necessity for importing water shows the considerable pressure on water resources and the delicate balance of water-human systems during droughts that calls for short-term and long-term sustainability
targets within drought policies.



## 1 Introduction

Groundwater plays a key role sustaining natural and anthropogenic water demand during meteorological droughts (De Graaf
et al., 2019; Siebert et al., 2010; Döll et al., 2012). Meteorological droughts, defined as periods of sustained dry weather
(Mishra and Singh, 2010), reduce water availability in soil moisture, surface water, and groundwater. Due to the natural delay
in groundwater recharge, it may take weeks, months, or even years before a precipitation deficit propagate through the hy-
drological cycle. Groundwater is available longer compared to surface water and is often used to complement water demand
during droughts (Taylor et al., 2013; Cuthbert et al., 2019). Increased groundwater use may result in aggravated streamflow
droughts, a deficit in discharge or reservoir storage (Mishra and Singh, 2010; Wada et al., 2013; Wanders and Wada, 2015).
Deficits in groundwater, caused by either absent recharge or increased groundwater use result in a groundwater drought defined
as a below normal groundwater level (Yevjevich, 1967; Tallaksen and Van Lanen, 2004). In this study, we focus on the human-
modified and human-induced hydrological droughts including surface water and groundwater use in below normal availability
of surface water or groundwater (Van Loon et al., 2016). Overexploitation of groundwater use, periodically during droughts
or permanently, may lead to groundwater depletion and reduced drought resilience (Custodio, 2002; Custodio et al., 2019).
Despite the important role of groundwater during droughts, the question remains how groundwater can be managed best and
whether drought management strategies can meet both environmental and anthropogenic water demand (White et al., 2019).

When national or regional drought policies are in place, water management during droughts is guided to structure drought
response and create drought resilience (Wilhite et al., 2014). Drought policies vary in their structure, focus on (different) water
users, and implementation. Key elements are 1) a drought definition, 2) monitoring of water resources and drought impacts, 3)
risk management, 4) (early) warning systems, 5) interventions or drought management strategies, 6) recovery and evaluation of
drought events (Wilhite et al., 2014; De Stefano et al., 2015; Urquijo et al., 2017). Studies aiming to compare drought policies
address these facets often in a qualitative manner for example when comparing Australia and the US (White et al., 2001;
Botterill and Hayes, 2012), different US states (Fu et al., 2013), and European countries (De Stefano et al., 2015; Urquijo et al.,
2017; Özerol, 2019). However, few of these drought policies are assessed in terms of their effectiveness (Urquijo et al., 2017;
Wilhite et al., 2014). In Europe, drought polices or drought management plans are evaluated as part of the Water Framework
Directive (abbreviated as WFD, EU Directive 2000) and member states are encouraged to move from crisis management
towards proactive management of droughts (Howarth, 2018). However, implemented drought policies vary (De Stefano et al.,
2015; Urquijo et al., 2017) and currently there is no consistent methodology to assess drought policies with respect to their
impact on water resources or hydrological droughts.

Methodologies to investigate feedback processes between water resource availability and drought management often use
socio-hydrological models to capture both hydrological and anthropogenic responses in time (Sivapalan et al., 2012; Di Bal-
dassarre et al., 2015). In this emerging field of applying socio-hydrological models to assess the impact of drought management,
most studies focus on one specific measure of a drought policy. For example, studies focused on maintaining environmental





flow requirements (Klaar et al., 2014), increased or altered groundwater use (Martínez-Santos et al., 2008; Apruv et al., 2017),
restrictions on water demand (White et al., 2019), conjunctive use of water resources (Huggins et al., 2018), management
regulations of reservoir storage (Di Baldassarre et al., 2018; Garcia et al., 2020; Dobson et al., 2020), or creating awareness of
water shortage during a drought (Garcia and Islam, 2019; Gonzales and Ajami, 2017). Jaeger et al. (2019) are the first to model
a set of drought policy measures. They tested drought measures separately and combined, showing that reservoir regulations
and timely interventions have a large impact on streamflow droughts. Alternative water sources, such as groundwater were
not considered. Given the importance of and increasing dependency on groundwater during drought (Aeschbach-Hertig and
Gleeson, 2012; Taylor et al., 2013; Cuthbert et al., 2019), there is a need to advance current drought policy modelling to include
policies that apply to both surface water and groundwater.

This study aims to assess the impact of drought policies on hydrological droughts and water resource availability for a
range of hydrogeological conditions. For this, we used a lumped socio-hydrological model to simulate drought management
strategies that apply to both surface water and groundwater. The socio-hydrological model represents an idealised (simplified)
hydrological system that includes a surface water reservoir, a groundwater module, and an option to import surface water. Environmental and anthropogenic water demand was met by withdrawing water from both surface water and groundwater stores.
Scenarios were used to evaluate separate and combined drought management strategies that altered proportional water demand,
source of water supply, and volume of imported surface water. These strategies were tested for a range of hydrogeological conditions (high, medium, and low groundwater storage systems) to assess their impact on different hydrological droughts and
water resource availability depending on virtual catchment settings.

## 2 Case study

To test and develop the socio-hydrological model, England is used as an case study considering the publicly available information on surface water and groundwater allocations during normal and drought conditions. Since 2003, water allocations
are based on a catchment water balance approach as WFD standards were integrated in national water policies (Environment
Agency, 2016; Howarth, 2018). Drinking water supply is the largest water user, comprising 55% of water demand on average
and up to 90% in some densely populated regions (data from 2000-2015 published by Environment Agency (2019a), presented
in A1). The privatised drinking water supply sector consists of 18 drinking water companies that provide drinking water in
England (Ohdedar, 2017; Ofwat, 2020). 13 out of the 18 companies use both surface water and groundwater, which water resource and drought management plans were used to inform baseline conditions and drought management scenarios (see Table
A1).

Water resource management plans show that the source of water supply varies depending on the regional variability of
surface water and groundwater. For example, companies with access to principal aquifers might depend more on groundwater
compared to companies with access to shallow, less productive aquifers (Table A1). In addition to locally available water, water
transfers between drinking water companies are regularly used to overcome seasonal or annual shortages. These transfers also
ease pressure on water resources and act as emergency supply during droughts (Dobson et al., 2020). The overall pressure on





water resources in the case study is considerable. During normal conditions the allocated water represents, on average, 88.5% of the available water that might increase during periods of high water demand or droughts (Table A1, Environment Agency 2019b). Not surprisingly, drought management plans are mandatory for drinking water companies to guide their drought
response. These plans are publicly available and often updated. Most recent plans were used in this study (see A2 for references to regional drought management plans).

Drought management plans consist of five main components: 1) drought definition, 2) warning system based on drought trigger levels, 3) demand management, 4) supply management, 5) evaluation of drought events (summarised in Table 1). Drought definitions and trigger levels are used to distinguish mild from severe drought events and activate management strategies with
increasing severity (Table 1). These drought trigger levels are often based on deficits in seasonal precipitation or the total precipitation in winter months (also called dry winters in drought management plans) that is the main groundwater recharge period in the UK. Water levels in rivers, reservoirs, and selected groundwater boreholes are also used as drought triggers when, for example, flow or storage levels are falling low. Drought management plans list various demand-related and supply-related drought management strategies that are activated for certain drought severity stages (see Table 1). Most commonly applied
strategies were implemented in the model (when permitted by the model setup) using the average effect of these measures, as reported in the drought management plans.

## 3  Model structure and data

The socio-hydrological model consists of a water balance model and a water demand model. The water balance model is driven by daily climate data that was selected to include the four most recent national hydrological drought events (Barker et al., 2019),
resulting in a period of investigation from 1980 to 2017. Based on this investigation period, a 5-year spin-up period was used to determine initial conditions (relevant to soil moisture and groundwater levels only). Note that this spin-up period includes water demand, but no drought management strategies. Baseline and drought management scenarios started thus in 1985 and ended in 2017. Natural (no water demand) model runs were used for reference purposes only (see Figure A3).

### 3.1  Model structure

The socio-hydrological model structure follows a standard conceptual water balance model with additional water demand components (Figure 1). The water balance model was based on the previously described lumped hydrological model of Van Lanen et al. 2013, who followed the widely applicable HBV model structure (Bergström, 1976). We extended the hydrological drought modelling regarding the groundwater module and water demand component. The hydrological model is driven by climate data, used as input for the soil moisture balance, generating runoff and groundwater recharge that are routed further to
the surface water reservoir and groundwater module, respectively. The water demand model is based on the regionally-averaged water resource and drought management plans, representing water management in the case study area.

The first model component is the soil moisture balance, represented by a medium soil (light silty loam soil: Soil II). The daily soil moisture balance ($SS$ for daily time steps $_t$ in mm) is determined by incoming precipitation (P in mm d$^{-1}$), actual





**Table 1.** Recent drought management plans of thirteen drinking water companies with staged drought management strategies according to drought trigger levels (see A2 for references to the drought plans). Average drought trigger levels are shown (range shown in square brackets) based on drought plans with trigger levels under 100 years for initial drought stages. Demand management and water supply strategies are shown per drought severity stage with modelled impact in $4th$ and $7th$ column respectively. Note that model scenarios are based on averaged reported effects when estimated (range of expected/reported impact is in parenthesis). Surface water and groundwater are abbreviated as SW and GW respectively for readability.

| Drought trigger level | Demand management plans | Applied in drought plans (#) | Modelled as | Supply management plans | Applied in drought plans (#) | Modelled as |
|---|---|---|---|---|---|---|
| Mild drought (1 in 8.5 year [5-20]) | Promote water use efficiency | 13 | Demand reduces | Maximise GW licence | 3 | GW use increases 4% (2-6%) |
| | Leak reduction | 13 | - | Import of SW | 10 | Water is imported when storage falls below 25% |
| | Water metering | 6 | - | Conjunctive use of SW & GW | 6 | Flexible use of SW & GW |
| | Temporary use ban (non-essential) | 13 | Demand reduces 5% (0-15%) | Maximise SW licence | 6 | SW use increases 6% (1-9%) |
| Moderate drought (1 in 22.5 year [10-80]) | Reduce pressure on water network | 7 | - | Deepening boreholes | 4 | - |
| | | | | River augmentation | 8 | - |
| | Temporary use ban (Commercial) | 12 | Demand reduces 12% (1-33%) | Reduce water export | 9 | - |
| | | | | Artificial recharge schemes | 1 | - |
| | | | | Reduction of ecological minimum flow | 8 | Ecological minimum flow not maintained |
| | | | | Maximise GW licence | 9 | GW use increases 7% (1-13%) |
| | | | | Maximise SW licence | 10 | SW use increases 14% (1-98%) |
| Severe drought (1 in 69 year [20-100]) | Phase winter & summer water use | 4 | - | Installation of additional GW wells | 6 | - |
| | Rota cuts | 8 | Demand reduces 36% (30-40%) | Reuse sewage water | 5 | - |
| | | | | Maximise GW licence | 10 | GW use increases 12% (1-49%) |
| | | | | Maximise SW licence | 9 | SW use increases 10% (2-26%) |





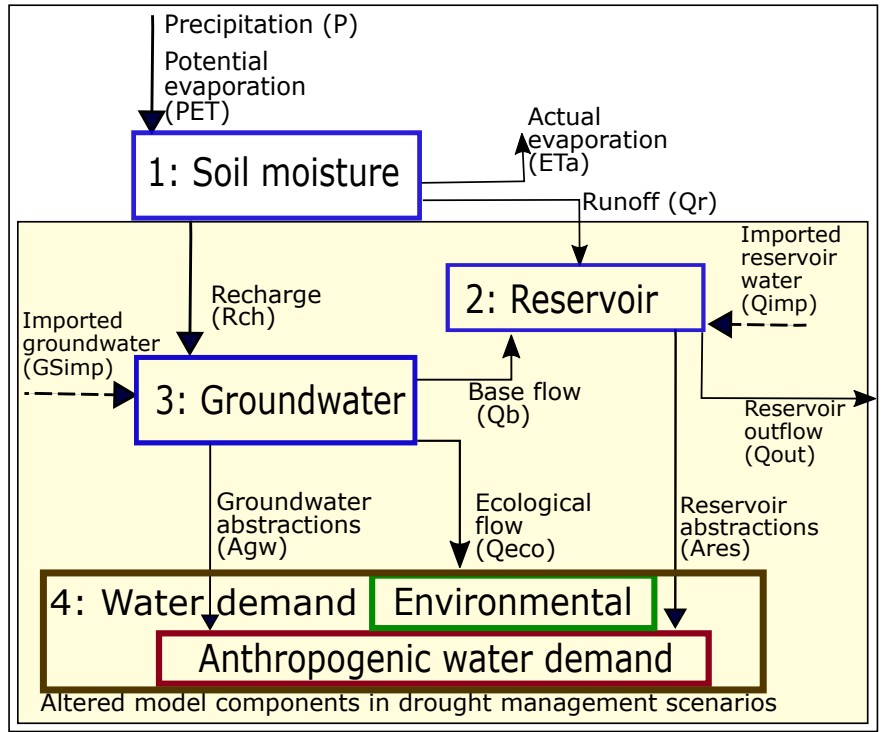

**Figure 1.** Socio-hydrological model consisting of a soil moisture balance (1) driven by precipitation (P in mm d$^{-1}$) and potential evaporation (PET in mm d$^{-1}$), a reservoir (2) storing runoff (Qr mm d$^{-1}$), and a groundwater module (3) driven by groundwater recharge (Rch in mm d$^{-1}$). Anthropogenic water demand (4) is met by reservoir (Ares in mm d$^{-1}$) and groundwater (Agw in mm d$^{-1}$) abstractions. Natural water demand is represented by ecological flow requirements (Qeco in mm d$^{-1}$) abstracted as part of the baseflow (Qb in mm d$^{-1}$). Remaining baseflow is routed to the reservoir. Additional water is imported in the model when reservoir or groundwater storage is insufficient (Qimp and GSimp both in mm d$^{-1}$). Drought management scenarios apply to the reservoir, groundwater module, and water demand (illustrated by the yellow box).

evaporation (ETa in mm d$^{-1}$) that was calculated from potential evaporation (PET in mm d$^{-1}$), overland flow or runoff (Qr in mm d$^{-1}$) and groundwater recharge (Rch in mm d$^{-1}$) (Van Lanen et al., 2013).

$$SS_t = SS_{t-1} + P_t - ETa_t - Qr_t - Rch_t \tag{1}$$

ETa was taken equal to PET when $SS_t$ is between field capacity ($SS_{FC}$) and critical soil moisture content ($SS_{CR}$), assuming that well-watered grass would in this case transpire at the potential rate. ETa was reduced for drier soils with a factor





$\frac{SS_t - SS_{WP}}{SS_{CR} - SS_{WP}}$, and below wilting point ($SS_{WP}$) ETa was assumed to be zero (Van Lanen et al., 2013). Qr occurs when the soil

reaches field capacity (168.9 mm) and when it is raining on very dry soil (below critical moisture content of 95.2 mm).

$$Qr_t \begin{cases} SS_t - SS_{FC} & \text{if } SS_t \geq SS_{FC} \\ 0 & \text{if } SS_{CR} < SS_t < SS_{FC} \\ \frac{1}{2}P & \text{if } SS_t \leq SS_{CR} \text{ \& P} > 2 \text{ mm d}^{-1} \end{cases} \tag{2}$$

Rch is calculated from the daily soil moisture content depending on the soil moisture retention shape parameter (b = 3 in average conditions; Seibert 2000) and the unsaturated hydraulic conductivity of Soil II ($k_{FC}$)Van Lanen et al. 2013; Tanji and Kielen 2002; Equation 3).

$$Rch_t = \begin{cases} 0 & \text{if } SS_t \geq SS_{FC} \\ \left(\frac{SS_t - SS_{CR}}{SS_{FC} - SS_{CR}}\right)^b k_{FC} & \text{if } SS_{CR} < SS_t < SS_{FC} \\ 0 & \text{otherwise } SS_t \leq SS_{CR} \end{cases} \tag{3}$$

The annual average runoff and groundwater recharge generated by the soil moisture balance also define the total available water for anthropogenic water demand, following the water resource management plans in the case study area. Allocated water is taken as a fraction (88.5%) of the total available water and divided equally over the days of the year (Table A1).

The second model component is a surface water reservoir storing runoff and baseflow (Figure 1). Stored water (in mm) is

used to meet the surface water demand, which is 44.6% of allocated water in the baseline and variable in the drought management scenarios. Maximum reservoir storage is set to one year of winter recharge, defined as the long-term total precipitation in the period December to February. Excess reservoir storage (Qout in mm d$^{-1}$ ) leaves the model and is not used to meet surface water demand. When storage declines, additional (unlimited) surface water (Qimp in mm d$^{-1}$) is imported in the baseline scenario. In drought management scenarios, reservoir storage is refilled when storage levels are below 25%, representing the

regular water transfers as part of the drought management strategies (see Table 1; also described in Dobson et al. 2020).

The third model component is the groundwater module that has three different options for hydrogeological conditions used for both baseline and drought management scenarios. These three options represent baseflow (Qb in mm d$^{-1}$) for different aquifer structures with high, medium, and low groundwater storage (GS in mm). Groundwater storage systems are based on the karstic, porous, and fractured aquifers in Stoelzle et al. (2015)). Baseflow generation is determined by different storage-

outflow parameter values ($s$ in d$^{-1}$) and different discharge represented by the different equations for three groundwater systems. Modelled values for $s$ are based on the range tested by Stoelzle et al. (2015) and cross-verified with mean karstic, porous, and fractured aquifer values as observed in England (Allen et al., 1997) (Table 2 with response time (in days) in parenthesis). Presented results are based on mean $s$ values (third row in Table 2) and alternative $s$ values were tested in the sensitivity analysis.





Aquifer structures for high, medium, and low groundwater storage are calculated using different equations based on the work of Stoelzle et al. (2015). The high groundwater storage system is modelled with a non-linear power law (Equation 4) using an average B value (0.5) based on the tested range of B values by Stoelzle et al. (2015).

$$\text{High groundwater storage system} = \begin{cases} Qb_t = sGS_t^B \\ GS_t = GS_{t-1} + Rch_t - Qb_t - Agw_t \end{cases} \tag{4}$$

The medium storage system is computed by a linear storage reservoir with additional by-pass component (D; Equation 5). The
by-pass component has a tested range of 0.07-0.12 by Stoelzle et al. (2015) and we used the average value (0.1) to allow 10% of groundwater recharge to by pass the groundwater system.

$$\text{Medium groundwater storage system} = \begin{cases} Qb_t = sGS_t + DRch_t \\ GS_t = GS_{t-1} + (1-D)Rch_t - Qb_t - Agw_t \end{cases} \tag{5}$$

The low storage system is represented by two parallel linear storage reservoirs with different storage-outflow parameters (Equation 6). The total groundwater storage is a sum of both parallel storage reservoirs, for which recharge and water demand
is equally divided.

$$\text{Low groundwater storage system} = \begin{cases} Qb_t = s_1GS1_t + s_2GS2_t \\ GS1_t = GS1_{t-1} + \frac{1}{2}Rch_t - s_1GS1_t - \frac{1}{2}Agw_t \\ GS2_t = GS2_{t-1} + \frac{1}{2}Rch_t - s_2GS2_t - \frac{1}{2}Agw_t \end{cases} \tag{6}$$

Groundwater abstractions (Agw in mm d$^{-1}$) were taken from the daily groundwater storage balance resulting in different time series for baseflow and groundwater storage for the three groundwater systems. From the generated baseflow, the ecological minimum flow (Qeco mm d$^{-1}$) is first withdrawn to allocate water for the environmental water demand. The remainder
of baseflow is routed to the reservoir and available for anthropogenic surface water demand. This implies that on days when baseflow is less or equal to Qeco, no baseflow is routed to the reservoir and all available water is allocated for environmental water demand, even though this might be less than the environmental flow requirements. Maintaining environmental flow requirements is only applied in some drought management scenarios, in which groundwater demand is restricted when flows fall below the ecological flow threshold. If groundwater storage is depleted, additional (unlimited) groundwater storage (GSimp in
mm d$^{-1}$ ) is imported to meet the groundwater demand that is additional to the water balance. In reality, additional groundwater would come from other aquifer sections, extending groundwater abstractions beyond the surface water catchment boundaries.

Hydrological drought characteristics were calculated applying a variable $80^{th}$ percentile of the baseline baseflow and groundwater time series corresponding to a 'once every 5 year drought' (Yevjevich, 1967; Tallaksen and Van Lanen, 2004; Mishra and Singh, 2010). This baseline threshold was also used for the drought management scenarios. In the sensitivity analysis,
where alternative storage-outflow parameters were tested, new drought thresholds were calculated taking the $80^{th}$ percentile of each baseline run (baseflow and groundwater storage time series) with an alternative parameters. Similar to the main analysis, impact of drought management strategies is computed from this baseline and new drought threshold.





**Table 2.** Groundwater storage-outflow $s$ values (in $d^{-1}$) for the three groundwater options in the groundwater module. The first row shows $s$ values used by Stoelzle et al. (2015), the second row shows representative $s$ values for England based on Allen et al. (1997), and the third row presents the modelled (mean) $s$ values for the three groundwater options in Equations 4-6. In the sensitivity analysis, a range of $s$ values was calculated (last row). For the low storage system, only $s_1$ was changed in the sensitivity analysis. The response time (in days) is shown for the modelled $s$ values in parenthesis.

| | High storage system ($s$ in $d^{-1}$) | Medium storage system ($s$ in $d^{-1}$) | Low storage system ($s$ in $d^{-1}$) |
|---|---|---|---|
| Optimal $s$ values by Stoelzle et al. (2014) | 0.008-0.025 | 0.001-0.01 | $s_1$ : 0.004-0.011<br>$s_2$ : 0.05-0.25 |
| Mean English $s$ values by Allen et al. (1997) | 0.009-0.04 | 0.0008-0.004 | 0.002-0.02 |
| Modelled $s$ values | 0.02 (50 days) | 0.004 (250 days) | $s_1$ : 0.005 (200 days)<br>$s_2$ : 0.1 (10 days) |
| Alternative $s$ values | 0.01 (100 days)<br>0.0133 (75 days)<br>0.03 (33 days) | 0.001 (1000 days)<br>0.002 (500 days)<br>0.01 (100 days) | 0.002 (500 days)<br>0.00285 (350 days)<br>0.01 (100 days) |

## 3.2 Data

Climate data for the hydrological model was selected to represent average climate conditions in England, providing an estimate
for precipitation (P) and reference potential evaporation (PET). Therefore, a regionally-weighted precipitation product was selected (at a daily time scale; Alexander and Jones 2001). In the absence of a regional (weighted) product for PET, a centroid location was selected to extracted daily time series from the (gridded) CHESS dataset of Robinson et al. 2016.

Water resource management plans were used to determine long-term (2000-2015) water demand and water availability for normal year (Environment Agency, 2019b). These documented water demand volumes were converted into a percentage (water
use divided by available water) representing water allocation per drinking water company (see Table A1). The average water allocation was 88.5% representing both surface water and groundwater demand. Between drinking water companies, water allocation varied between 82% and 95% (Table A1), which was further explored in the sensitivity analysis by in/decreasing water allocation with 5% (to 93.5% and 83.5% respectively). The proportions of surface water and groundwater also varied between companies and an average was used for surface water (44.6%) and groundwater (48.5%) demand. The remaining water
demand (6.9%) was provided by imported water representing water transfers between companies during normal conditions and during droughts (Dobson et al., 2020). Considering the large range of surface water and groundwater demand between the





companies (15-88% and 10-84%, respectively), alternative proportions of surface water and groundwater demand were tested
in the scenarios.

Data from the regionally-averaged drought management plans was used to define drought trigger levels and activate drought
management strategies (Table 1). Modelled trigger levels were based on averaged reported drought trigger levels, excluding ex-
tremely long return periods (100-150 year) for initial drought stages. Trigger levels are applied to precipitation (using monthly
SPI), streamflow, and groundwater level time series, as is common for the drinking water companies. For example, if either
surface water or groundwater falls below the trigger level, for example, in a 1 in 8.5 year drought event, the first category of
drought management strategies will be activated. Different trigger levels are applied to reservoir storage levels that are kept

relatively full with a 30-60 day emergency storage. Reservoir trigger levels in the first drought category typically start from
80% to 60% of reservoir storage, second category from 60% to 30%, and the last from 30% to 12%. These percentages are
converted to reservoir trigger levels of 75%, 50%, and 25%.

Based on the listed drought management strategies, four scenarios were developed testing first four separate strategies (Table
3). The first scenario focuses on water supply and includes an increase in water demand for both surface water and groundwater

based on the reported range in Table 1. The second scenario focused on restricting water demand and reduces surface water
and groundwater demand based on reported (achieved or modelled) water demand reductions (Table 1). The third scenario
introduced conjunctive water use as a drought management strategy that integrates surface water and groundwater demand.
Daily water demand is provided by either water source depending on the highest available storage. The fourth scenario main-
tains ecological flow requirements (also known as 'hands off flow'). Environmental water demand is maintained by restricting

groundwater demand when baseflow falls below the seasonal ecological minimum flow threshold ($80^{th}$ percentage). In addition
to these four separate drought management strategy scenarios, two combined scenarios were tested to investigate the combined
effect of gradual in/decrease of water demand with either conjunctive use (scenario 'combined 1-2-3'), or maintaining the
ecological flow (scenario 'combined 1-2-4').

**Table 3.** Description of rules applicable to the four separate drought management strategy scenarios. Note that staged drought management
strategies under the first and second scenario (1: Water supply and 2: Restricted use) are activated by drought trigger levels. The third and
fourth scenario are active throughout the modelling period. Modelled scenarios are based on (averaged) documented drought management
strategies, see Table 1 for details.

|  | 1: Water supply | 2: Restricted use | 3: Conjunctive use | 4: Maintaining ecological flow |
|---|---|---|---|---|
| Mild drought | + 6% surface water demand<br>+ 4% groundwater demand | Water demand -5% | Integrated surface<br>water and groundwater<br>storage use | No groundwater use,<br>when baseflow falls below<br>ecological minimum flow |
| Moderate drought | + 14% surface water demand<br>+ 7% groundwater demand | Water demand -12% | | |
| Severe drought | + 10% surface water demand<br>+ 12% groundwater demand | Water demand -36% | | |
| Applicable at all times: | Surface water import when reservoir levels fall below 25% | | | |





## 4 Results

The results are presented in four sections starting with baseline conditions for the three modelled hydrogeological conditions. Next, drought management scenarios are presented and their impact on hydrological droughts is shown relative to the baseline. The sensitivity analysis with alternative groundwater-outflow parameters and baseline water demand is presented last.

### 4.1 Baseline

In the baseline scenario, the soil moisture balance shows inter-annual variations, but no systematic wetting or drying, as the total
water balance is close to zero (18mm) for 37 years (see Figure A2). Periods of below-normal precipitation resulting in reduced groundwater recharge and runoff are visible in spring 1989, 1991-1992, 1996-1997, 2003-2004, 2005-2006, 2010-2012, and June 2017. These periods are colour-coded according to drought definitions in Table 1 in Figure 2. Periods of above-normal precipitation are noted in 1991, 2001 and 2012 resulting in a saturated soil with excess runoff generation instead of recharge.

Reservoir storage in the baseline follows the inter-annual variability in runoff and baseflow that is generated by the ground-
water module (Figure 2). Reservoir storage is lowest in the high groundwater storage system (mean: 16%, range: 0-89%). In the medium and low groundwater storage systems, surface water storage levels are higher with on average 36% and 66% reservoir storage, respectively. In the low groundwater storage system, low reservoir levels occur during mild droughts only. When reservoir storage declines, additional surface water is imported to meet the daily surface water demand. This additional import represents 8.1%, 1.7%, and 0.3% of the total water demand for the high, medium, and low groundwater storage sys-
tems, respectively (Figure 3). The proportions of additional surface water imports are considered within the range of common in/exports of surface water in England (see A1).

Groundwater availability is highest in the high groundwater storage system and smaller for the other two systems (medium and low groundwater storage systems; Figure 2). Groundwater storage in the high storage system buffers more mild droughts compared to the other two systems, for which groundwater storage depletes rapidly in summer months resulting in lower base-
flow and ecological flow requirements in these systems. Compared to scenarios without water demand (Figure A3), groundwater storage and baseflow are much lower, showing the pressure on groundwater systems given the current anthropogenic groundwater demand. The required additional groundwater import to meet the daily groundwater abstractions represents a relatively small proportion of the total water demand (1%) in the high groundwater storage system. In the medium and low systems this share is larger (11% and 17% respectively; see Figure 3). Considering the similarity in results for the medium and
low groundwater storage systems in surface water and groundwater availability, results for the drought management scenarios are only shown for the high and low groundwater storage systems.

### 4.2 Drought management scenarios

Out of the four drought management scenarios, conjunctive use of surface water and groundwater has the largest impact on surface water and groundwater availability in the high and low groundwater storage system (Figure 4). Results of the medium
groundwater storage system are not shown as results are very similar to the low groundwater storage system. In the conjunctive

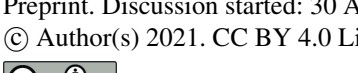


**Figure 2.** First panel shows the standardised Precipitation Index (SPI) for regionally averaged *monthly* precipitation. Drought severity is indicated in three colours according to three drought stages in drought management plans (Table 1). Other three panels show *daily* baseline conditions for reservoir storage and groundwater availability for high (green), medium (gold), and low (blue) groundwater storage systems. Note that y-axes are different for the three systems. Reservoir capacity is defined as the total long-term winter precipitation and therefore constant in the three systems.


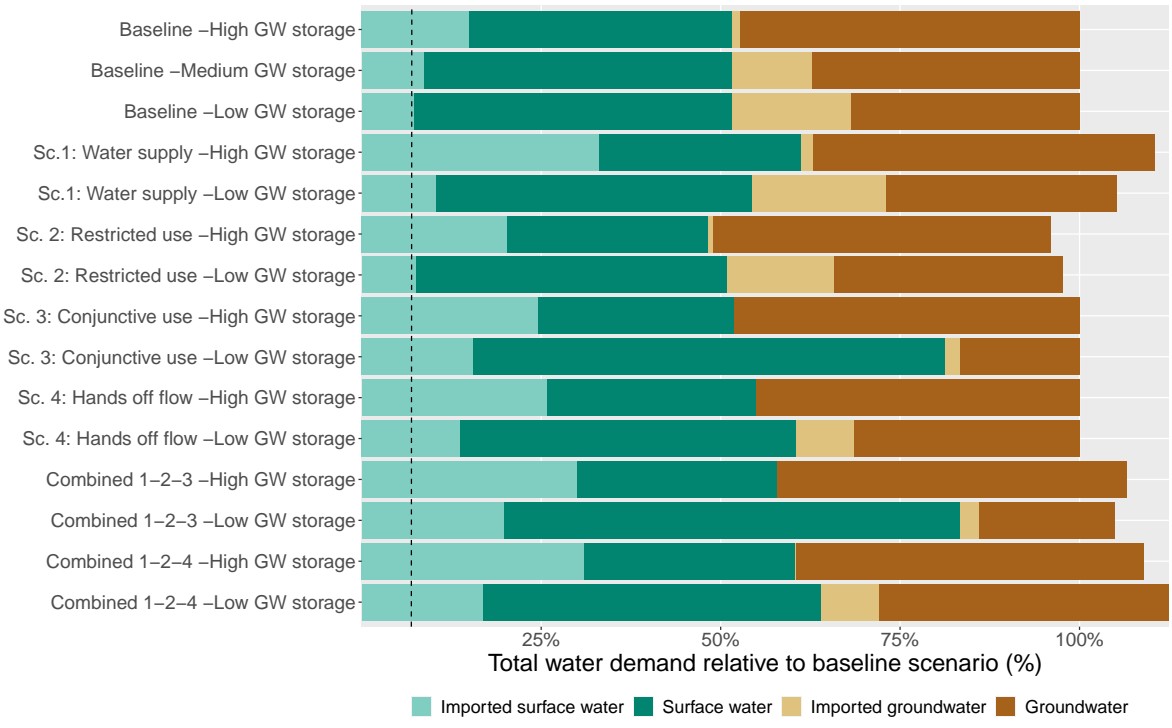

**Figure 3.** Total water demand for the baseline scenario for the three groundwater storage systems (rows 1-3). Total water demand is met by a combination of surface water (imported and in reservoir) and groundwater (imported and locally available). The constant surface water import of 6.9% of the total anthropogenic water demand is indicated by the dotted vertical line. Separate drought management scenarios (rows 4-11) and combined scenarios (12-15) are shown for the high and low groundwater storage systems only. Note that total water demand in scenarios can be different to baseline conditions due to the drought management strategies and that 100% refers to the total water demand in the baseline. Names of both groundwater storage systems are abbreviated as 'High/Low GW storage' for readability.

use scenario, surface water and groundwater use are integrated meeting the overall water demand resulting in flexible water demand. In the low groundwater storage system, reservoir storage is used more intensively representing 65.6% of total water demand (Figure 3). Applying conjunctive water use increases groundwater storage, as groundwater use decreases to 17% resulting in a 50% increase in baseflow compared to the baseline. In the high groundwater storage system, surface water and

groundwater use change mainly in timing and show a minimal change in proportional surface water and groundwater use compared to the baseline (Figure 3). Baseflow remains high, similar to the baseline, although groundwater storage reduces slightly (Figure 4). Additional groundwater import reduces to a minimum in both systems, although this comes at the expense of imported surface water, which increases with 9.6% and 8.3% to 24.5% and 15.5% in the high and low groundwater storage systems respectively (Figure 3).

Second to the conjunctive use scenario, the fourth scenario 'hands off flow' also has substantial impact on the high ground-water storage system resulting in higher groundwater storage and baseflow (on average 14%; groundwater time series shown



**Figure 4.** Impact on groundwater storage following from the four separate drought management scenarios. Coloured surfaces match the increasing severity of meteorological droughts (related to trigger levels, see Table 1). Baseline conditions for high and low groundwater storage systems are shown in the first and third panel. Second and fourth panel show the impact on storage (baseline minus scenario). Applied rules for the four separate drought management strategies are presented in Table 2.

in Figure 4). The restrictive use of groundwater to maintain ecological minimum flow requirements results in a continuous increase in groundwater storage in the high storage system, compared to periodic increases in storage in the low storage system. The periodically increasing groundwater storage results in a small increase in baseflow (on average 1%) suggesting that this scenario has much less impact in the low groundwater storage system. With the restricted use of groundwater, surface water demand increases 2.2% to meet the anthropogenic water demand. Consequently, imported surface water increases 6.5% in


the low storage system. In the high storage system, reservoir storage is already optimised and a larger proportion of imported surface water (additional 10.7%) is used to meet the remaining anthropogenic water demand (Figure 3).

The first and second scenarios that introduce drought mitigation strategies during meteorological droughts result in periodic
in/decreases of surface water and groundwater storage (Figure 4). The first scenario that increases water supply during droughts results in small storage deficits that recover after the drought events. The second scenario introducing reductions in water demand shows a similar, but opposite, pattern with increasing groundwater storage during most severe meteorological droughts caused by the severe restrictions on water demand. Compared to the baseline, water restrictions in the second scenario reduce the overall water demand slightly for high and low storage system (96% and 98%, respectively; Figure 3). The impact of the
first scenario (increased water supply) is larger, as the total water demand exceeds the baseline water demand with 11% and 5% respectively for high and low groundwater storage systems due to increased surface water import (Figure 3).

The two combined drought management scenarios show an overall increase in baseflow and groundwater storage. Combining conjunctive use with scenarios 1 and 2 (combined 1-2-3 scenario) increases groundwater storage in the low groundwater system resulting in higher baseflow of 42% on average. Groundwater storage reduces slightly in the high storage system, but baseflow
remains high. For the high storage system in particular, combining 'hands off flow' with scenarios 1 and 2 (combined 1-2-4 scenario) increases baseflow up to 14% compared to only a 1% increase in the storage low system. Both combined scenarios result in a slightly higher total water demand compared to baseline due to increased water supply during droughts in scenario 1. However, the total water demand is lower compared to scenario 1 implying that water demand restrictions (scenario 2) compensate for additional water supply during droughts. The use of imported groundwater reduces in both combined scenarios,
but the dependency on imported surface water increases, which is related to import of surface water as reservoir levels fall below 25% (Table 3). This is because, reservoir triggers are activated during most meteorological droughts importing surface water to complement low reservoir levels (time series of reservoir levels in Figure A4).

### 4.3   Impact on hydrological droughts

In the baseline, there is a large difference in hydrological drought characteristics between the two groundwater storage systems
(Table 4). Baseline conditions show longer baseflow and groundwater droughts (on average 333 and 344 days) in the high groundwater storage system compared to shorter hydrological droughts in the low storage system (66 and 88 days for baseflow and groundwater). The shorter hydrological droughts are remarkably intense resulting in no flow or extremely low storage levels with a rapid recovery during winter months and an overall flashy time series for both baseflow and groundwater in the low groundwater storage system (Figure 5). When winter recharge is low, high drought intensities are found compared
to hydrological drought intensity of the high groundwater storage system. Due to the higher storage component, precipitation deficits have a longer propagation with consequently fewer, more intense hydrological droughts. The low groundwater storage system is on the other end of the spectrum with double the amount of groundwater droughts compared to meteorological droughts. Given the different drought characteristics in the high and low groundwater storage systems, the impact of drought management strategies (separately or combined) is also variable and sensitive to the primary groundwater storage conditions.





**Table 4.** Hydrological drought duration, maximum intensity, and drought frequency for the high and low groundwater storage systems. Mean hydrological (baseflow and groundwater) droughts are presented for baseline, combined 1-2-3, and combined 1-2-4 scenarios. See Table 3 for specific drought strategies in these scenarios. Groundwater storage time series and groundwater droughts are shown in Figure 5.

| | | Drought duration (in days) | | Maximum drought intensity (in mm) | | Drought frequency (count of events) | |
|---|---|---|---|---|---|---|---|
| | | Baseflow | Groundwater | Baseflow | Groundwater | Baseflow | Groundwater |
| High groundwater storage system | Baseline scenario | 333 | 344 | -0.16 | -96.2 | 7 | 7 |
| | Combined 1-2-3 scenario | 145 | 152 | -0.04 | -51.7 | 24 | 23 |
| | Combined 1-2-4 scenario | 165 | 166 | -0.04 | -45.1 | 6 | 6 |
| Low groundwater storage system | Baseline scenario | 66 | 88 | -0.31 | -16.0 | 25 | 20 |
| | Combined 1-2-3 scenario | 58 | 62 | -0.38 | -14.3 | 8 | 5 |
| | Combined 1-2-4 scenario | 67 | 92 | -0.32 | -18.2 | 20 | 15 |

In the combined scenario including conjunctive use (combined 1-2-3), groundwater droughts are shorter in both systems compared to baseline conditions (Table 4). Hydrological drought intensities reduce in the high groundwater storage system, compared to a slight increase in baseflow droughts in the low storage system. Drought frequencies of both baseflow and groundwater show a sharp contrast between the two systems, as drought frequency increases from 7 events to 24 and 23 for baseflow and groundwater in the high storage system, compared to a reduction in hydrological droughts in the low storage

system. Groundwater time series in the low storage system in Figure 5 show that short groundwater droughts are alleviated and remaining events are of a shorter duration and reduced intensity. However, in the high storage system, hydrological drought frequency increases. Drought events occur without initial precipitation deficits, which might be related to the altered reservoir and groundwater abstractions.

     The combined scenario including hands off flow (combined 1-2-4) also shows mixed impacts on hydrological droughts in the

two systems. In the high groundwater storage system, drought intensity and duration reduce on average compared to baseline conditions (Table 4). Time series show alleviated groundwater droughts in 1993 and 2009 (Figure 5). In the low storage system, however, the impact of the 1-2-4 combined scenario is much lower with a slight reduction in drought intensity and duration. This is not surprising considering the overall low ecological minimum flow and respectively limited impact with introducing groundwater use restrictions.



**Figure 5.** Hydrological droughts shown for the baseline scenario and the six tested drought management scenarios (four separate scenarios and two combined scenarios). In the first and third panel, time series of groundwater level variation in the two groundwater storage systems (high and low) are shown for both baseline (black) and combined scenarios (combined 1-2-3 in dotted blue and combined 1-2-4 in striped red). Baseline drought events are marked in grey following the drought threshold (grey striped). Coloured surfaces indicate mild, moderate, and severe meteorological droughts (measured in SPI) following definitions in Table 1 and colour scale of Figure 2. In the second and fourth panel, groundwater drought occurrence and maximum intensity is shown for drought management scenarios for both catchments. Note that the coloured maximum drought intensity scale is the same for both catchments with red being the most severe and blue representing least intense droughts.





## 4.4  Sensitivity analysis

The sensitivity analysis aims to test averaged model parameters considering the large range reported in the case study. The first tested parameter is the groundwater storage-outflow parameter using a wide range of parameters based on previous work relevant to the case study (Allen et al., 1997) and modelling work (Stoelzle et al., 2015). The second tested parameter examines the large range of overall water demand based on the reported range by drinking water companies (A1). Other parameters in the water balance model were not changed from widely applicable HBV model structure (Bergström, 1976) or the hydrological drought model by Van Lanen et al. (2013).

### 4.4.1  Groundwater storage-outflow parameters

Alternative groundwater storage-outflow parameters are based on aquifer characteristics in England (Allen et al., 1997) and the range of optimal groundwater storage-outflow coefficients by Stoelzle et al. (2015) (parameters are shown in Table 2). These sensitivity tests show that the absolute groundwater storage in the high groundwater storage system is highly sensitive compared to the low groundwater storage system (Figure A5). However, this sensitivity has limited consequences for hydrological droughts, as drought duration and intensity increase slightly for each drought event (Figure 6). In the low groundwater system, for which the absolute change in storage is small, hydrological drought duration nearly double. Maximum hydrological drought duration increase from 137 days (baseflow) and 237 days (groundwater), to 273 and 455 days, respectively. These droughts also increase slightly in intensity, but much less compared to the drought duration (Figure A5).

When running the drought management scenarios (combined scenarios only) with these different groundwater storage-outflow parameters, the overall hydrological drought intensity and duration reduce for most scenarios (see Figure A6). The combined scenario 1-2-4 (including maintaining the ecological minimum flow) reduces hydrological drought duration for all groundwater storage-outflow parameters, even for high storage parameters in the two different groundwater storage systems (Figure A6). The combined scenario 1-2-3 (including conjunctive use) results in longer droughts, but less severe droughts, particularly for increased storage parameters in the low groundwater storage system. In the high groundwater system, groundwater drought duration increases dramatically with the highest groundwater storage parameter, as groundwater storage declines in this scenario and falls below the drought threshold resulting in a depleted system with exceptionally long drought.

### 4.4.2  Overall water demand

Altering the overall water demand by 5% shows the sensitivity to increasing pressure on water resources resulting in lengthened droughts in the high groundwater storage system and an increase in surface water import. When increasing the water demand (from 88.5% to 93.5%), hydrological drought duration in the high groundwater storage system lengthens up to 866 and 867 days for baseflow and groundwater respectively (Figure 6). This is nearly doubling hydrological drought duration in the baseline (Table 4). Increased water demand results also in additional shorter events that increase the drought frequency. Reducing water demand by 5% results in fewer severe droughts (Figure 6). This drought alleviation would, however, require a permanent cut in water consumption in addition to the introduced water restrictions during drought events. In the low groundwater storage system





is much less sensitive to in/decreasing water demand, as drought duration and severity are similar to the baseline. However, drought characteristics might not show the impact of altered water demand, as these tests mainly change the proportion of imported groundwater and surface water.

When testing the total water demand with the combined scenarios, the primary findings is an increase in imported surface water and groundwater. Both combined drought scenarios reduce hydrological droughts successfully (Figure A7), although this comes at the cost of increased surface water and groundwater imports. For example, increased water demand (93.5%) in the high groundwater storage system with the combined 1-2-4 scenario reduces maximum hydrological drought duration from 866 and 867 days to 308 and 309 days for baseflow and groundwater, respectively (Figure A7). This drought alleviation
comes with an increase of imported surface water representing up to 30% of the total increased water demand. Reduced water demand (83.5%) results in shorter droughts of maximum 218 days with slightly less surface water import (27% of total water demand). These increased percentages of imported surface water show the pressure on water resources and true cost to reducing hydrological droughts in combined drought management scenarios.

## 5   Discussion

### 5.1   Model

In this study, the impact of drought management strategies on hydrological droughts was investigated using a socio-hydrological model for a range of hydrogeological conditions. Comparing different drought management strategies in a quantitative manner, as presented here, complements qualitative comparisons of previous studies (White et al., 2001; Wilhite et al., 2014; Urquijo et al., 2017). Some of the tested strategies have been assessed separately, as studies focused on either water demand (Low
et al., 2015; Maggioni, 2015; Gonzales and Ajami, 2017; Hayden and Tsvetanov, 2019), adaptive water management (Thomas, 2019; White et al., 2019), or conjunctive use combined with managed aquifer recharge to increase drought resilience (Scanlon et al., 2016; Alam et al., 2020). Jaeger et al. (2019) and Dobson et al. (2020) show that combined drought policy interventions mitigated streamflow droughts by altering reservoir storage regulations and transfers. Results in this study agree with these findings showing reduced baseflow droughts in combined and separate drought management scenarios, but important
differences are found between the tested hydrogeological conditions. When integrating both reservoir and groundwater storage by applying conjunctive use in a low groundwater storage system, baseflow increases and hydrological droughts reduce. This comes, however, at the expense of additional surface water import that fulfills storage deficits in groundwater. Even though water is regularly traded between water companies (Dobson et al., 2020), percentages exceeding 10% of the total water demand are uncommon (see A1 for normal conditions). In high groundwater storage systems, conjunctive use reduces the intensity of
hydrological droughts, but restricted groundwater use during low flow periods proves to be most effective in reducing hydrological droughts when additional surface water imports are available.

The different response to drought management strategies is also related to the different drought characteristics of the high and low groundwater storage systems. These hydrogeological conditions show a positive relation between drought duration and groundwater-outflow storage properties confirming earlier studies in natural settings using a virtual model (Van Lanen

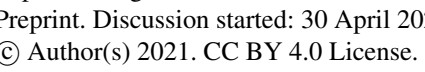

**Figure 6.** Impact of in/decrease modelled storage-outflow parameters and in/decreased water demand on groundwater drought characteristics (drought duration and maximum intensity). The range and reference for tested groundwater storage-outflow parameters can be found in Table 2. The range of documented water allocation of the selected drinking water companies can be found in A1. The first two panels show drought characteristics of the high groundwater storage system. The second two panels represents drought characteristics for the low groundwater storage system. Drought impacts following mean values for storage-outflow parameters and water allocation are shown in squares (all panels).





et al., 2013; Van Loon et al., 2014) and a spatially-distributed model (Carlier et al., 2019). Hydrological droughts in the high groundwater storage system are longer and have a longer drought recovery. In the low groundwater storage system, mostly short climate-controlled droughts are observed, which was also found by Stoelzle et al. (2015). Both baseflow and groundwater droughts have a short response time and limited lengthening of hydrological droughts even when the pressure on water resources increases. These findings match observations made across English aquifers that are characterised by a low or

high groundwater storage component (Bloomfield and Marchant, 2013; Bloomfield et al., 2015).

### 5.2  Impact of drought management strategies on hydrological droughts

Out of the four separate drought management strategies conjunctive use is most effective in easing pressure on water resources resulting in reduced hydrological droughts, increased baseflow and groundwater storage, particularly in the low groundwater storage system. Scenarios show the potential of integrating both water resources, as management strategy resulting in increased

drought resilience (Scanlon et al., 2016; Noorduijn et al., 2019; Holley et al., 2016). However, conjunctive use does not create water, but optimises storage use, particularly in catchments with large reservoir storage (Bredehoeft, 2011). Flexible use of surface water and groundwater aligns the timing problem between water demand and availability (Taylor et al., 2013; Cuthbert et al., 2019). It should also be noted that conjunctive use could also alter the river regime (not tested due to model setup), resulting in adverse impacts on ecohydrology (Rolls et al., 2012). We observed altered groundwater storage patterns in the

high groundwater storage system, resulting in lower groundwater storage with more frequent, but less intense hydrological droughts with potential severe consequences for longer meteorological droughts. This was also found by Shepley et al. (2009), who found that groundwater levels fell due to increased groundwater use in an English conjunctive use system. Optimising the timing of surface water and groundwater use seems key for a successful conjunctive system, although the required flexibility might have practical limitations for water managers (Bredehoeft, 2011). For example, water use licences are often set to a

specific water source and re-allocation of water licences can be difficult, which limits implementation of conjunctive use (Holley et al., 2016). However, a degree of flexibility can be achieved when water management units are large enough to contain multiple source-specific licences (Shepley et al., 2009; Fowler et al., 2007; Thorne et al., 2003).

Maintaining the ecological minimum flow requirements is also very effective in mitigating hydrological droughts, particularly in the high groundwater storage system. This confirms earlier findings focusing on the protection of ecosystems using

trigger level regulations (Werner et al., 2011; Noorduijn et al., 2019). Crucial to the success is the integration of surface water and groundwater use to maintain low flows (Howarth, 2018). However, results show that impact of restricting groundwater use during low flows relies on the defined trigger level (defined ecological minimum flow) and baseflow component, as protecting the minimum flow might not preserve natural or undisturbed river flows (Howarth, 2018). When increasing storage-outflow parameters in the sensitivity analysis and thereby increasing the baseflow component, impact of restricting groundwater use

increases. Crucially, hydrological droughts aggravate when the ecological minimum flow is neglected and groundwater use reduces the environmental flow (Gleeson and Richter, 2018; De Graaf et al., 2019). These crucial sensitivities to different groundwater-outflow parameters show the value of conceptual socio-hydrological modelling, which outcomes could be used





in the discussion regarding the protection of groundwater dependant ecosystems and the status of protected water bodies (Ohdedar, 2017; Howarth, 2018).

Combined drought management strategies show primarily the impact of conjunctive use and restricted groundwater use in both systems. The impact of drought mitigation scenarios 1 and 2 (increased water supply and restricted water demand) is mostly noticeable during extreme drought conditions when water demand reduces more than water supply increases. In most extreme drought conditions, water demand reduces by 36% that is similar to extreme water reductions realised in Melbourne during the Millennium Drought (Low et al., 2015), but not as low as water restrictions enforced in some parts of Cape Town

during the Day Zero crisis (Rodina, 2019; Garcia et al., 2020).

    When introducing a permanent increase in water demand (+5%), the effect on water resources is evident as hydrological droughts increase disproportionally in duration and required additional surface water import to meet the anthropogenic water demand. Further research is required to assess if these volumes of imported water are obtainable during droughts, especially considering the scale of drought events and potentially limited water availability at regional or even national scales. Reducing

water demand (-5%) results in shorter hydrological droughts and less imported water, but realising a permanent reduction in water demand can come at high costs for both providers and users, and might not always be successful (Low et al., 2015; Gonzales and Ajami, 2017; Muller, 2018; Caball and Malekpour, 2019; Simpson et al., 2019). Generating more awareness and reducing water demand prior to the actual water shortage might also result in better adaptive management of water resources (Garcia et al., 2016; Noorduijn et al., 2019; Garcia et al., 2020; Thomann et al., 2020).

## 5.3   Model limitations

Limitations of the conceptual socio-hydrological model are related to the overall drawbacks of using a lumped and idealised hydrological model. The regionally-averaged model input for both climate time series and water management means that model outcomes are generic and broadly representative for water resource availability in an English setting. Model runs to determine water availability and drought impact for specific regions in England would therefore require different climate data

and additional information regarding local water resource and drought management practices.

    The lumped model structure reduced testing of some drought management strategies that would require a spatially-distributed model. Out of the listed strategies (Table 1), four drought scenarios were tested in this study. Other measures, such as river augmentation (groundwater abstraction to supplement river flow during drought), reduction of pressure on the water network, and reuse of urban wastewater could not be modelled. A spatially-distributed setup could further the current analysis, as spatial

impact of increased abstractions to the stream could not be included (Gleeson and Richter, 2018) that would be relevant to the estimate the regional impact on hydrological droughts of scenarios applying conjunctive use or maintaining ecological flow requirements.

    If more water demand or water management data were available, current assumptions could be improved. For example, the static water demand could be substituted by a dynamic water demand component or increased awareness of water stress (Garcia

et al., 2016), if this would be supported by water resource or drought management plans. Conjunctive use scenarios could also



benefit from additional information regarding general water management practices, as practical constrains to flexible water storage can limit the effectiveness of conjunctive use (Holley et al., 2016).

## 6 Conclusions

This study presents a socio-hydrological model that was used to investigate the impact of water demand and drought management strategies on hydrological droughts. In the socio-hydrological model, different groundwater storage systems were modelled revealing different drought characteristics and impact of drought management strategies on hydrological droughts. Baseline conditions show that hydrological droughts occurred frequently and were mostly climate-driven, although amplified by water use in the low groundwater storage system. External water imports were necessary to meet water demand periodically. The high groundwater storage system shows larger inter-annual storage resulting in fewer, but more intense hydrological droughts amplified by water use.

Introducing drought management strategies to the different groundwater storage systems relieved both streamflow and groundwater droughts in nearly all scenarios. Most hydrological droughts are alleviated when applying conjunctive use and maintaining the ecological flow requirements by restricting groundwater use. The conjunctive use scenario allowed a more optimal use of reservoir storage and delayed response of groundwater storage resulting in reduced and sometimes alleviated streamflow droughts in the low and high groundwater storage systems. These findings encourage further exploration of conjunctive use as a drought mitigation strategy, particularly in low groundwater storage systems. The impact the restricted groundwater use to maintain ecological flow requirements (hands off flow) was found sensitive to the baseflow component, as hydrological droughts are effectively reduced under a range of storage-outflow parameters and when overall water demand was in/decreased.

The novelty of this study lies in the introduction of the socio-hydrological model to assess of the impact of drought management strategies on both streamflow and groundwater droughts. Results show how strategies as conjunctive use and maintaining ecological flow requirements reduce and alleviate hydrological droughts. The low sensitivity of these drought management strategies to different hydrogeological conditions highlights the wide applicability of results and gives confidence in the tested combined and separate scenarios. However, the considerable pressure on water resources is evident when the overall water demand increased, as drought duration increases disproportionally and additional surface water is required to meet the anthropogenic water demand. Further conceptual modelling could investigate the introduced dependency on imported water with these drought management strategies. The necessity for importing water shows the considerable pressure on water resources and the delicate balance of water-human systems during droughts that calls for sustainability targets within drought policies.

*Code availability.* Code available on request



*Data availability.*  Input data for the case study is freely available. Regionally averaged precipitation data can be found on the Met office Hadley Centre (website: https://www.metoffice.gov.uk/hadobs/hadukp/). Spatially-distributed data can be found on the UK water resources portal (website: https://nrfa.ceh.ac.uk/content/uk-water-resources-portal). Information about water resource and drought management plans is also publicly available and used plans are listed in A2.





**Appendix A: Supplementary material**

**A1   Water use and sources of water supply for drinking water companies in England**

**Table A1.** Summary of characteristics of drinking water company that use both surface water and groundwater in England. Drinking water companies South West and Northumbrian water are therefore excluded form this overview. Data of latest water resource management plans has been used (see A2 for source web-locations). Imported and exported percentages are marked with an asterisk when the source was undefined (or potentially mixed). Thames Water values shown for both London and outer areas in parenthesis. Headroom is calculated taking reported baseline conditions demand: supply (dated in 2019/20) and checked with published data of Environment Agency (2019b).

| Drinking water company | Supplies to # million customers | Surface water (%) | Groundwater (%) | Imported water (%) | Headroom (%) |
|---|---|---|---|---|---|
| Affinity Water | 3.6 | 28 | 65 | 7 | 86 |
| Anglian Water | 6 | 41 | 50 | 9 | 86 |
| Bristol Water | 1.2 | 42 | 12 | 42 | 93 |
| Portsmouth Water | 0.7 | 35 | 55 | 10 | 94 |
| Severn Trent Water | 8 | 67 | 33 | - | 92 |
| South East Water | 2.2 | 28.5 | 70 | 1.5 | 83 |
| Southern Water | 2.3 | 22 | 70 | 8 | 82 |
| South Staffs Water | 1.3 | 60 | 40 | - | 95 |
| Sutton & East Surrey Water | 0.7 | 15 | 84 | 1* | 84 |
| Thames Water | 15 | 80 (25) | 20 (70) | - (5) | 91 |
| United Utilities | 3 | 88 | 10 | 2 | 94 |
| Wessex Water | 2.8 | 21 | 75 | 4 | 88 |
| Yorkshire Water | 2.3 | 71 | 25 | 4 | 83 |
| **Average** | 3.8 | 44.6 | 48.5 | 6.7 | 88.5 |





## A2   Drought management plans of drinking water companies

**Table A2.** Locations of drought management plans of twelve drinking water company in England. All drought management plans are publicly available (websites are stated in second column). Most recent date is shown in third column with the last access date.

| Drinking water company | Drought management plan | Dated at | Last accessed |
| --- | --- | --- | --- |
| Affinity Water | affinitywater.co.uk/drought-management | 2018 | 2-9-2020 |
| Anglian Water | anglianwater.co.uk/drought-plan | 2019 | 2-9-2020 |
| Bristol Water | bristolwater.co.uk/planning-for-drought | 2018 | 2-9-2020 |
| Portsmouth Water | portsmouthwater.co.uk/final-drought-plan-2019 | 2019 | 2-9-2020 |
| Severn Trent Water | severntrent.com/our-plans | 2019 | 2-9-2020 |
| South East Water | corporate.southeastwater.co.uk/drought-plans | 2019 | 2-9-2020 |
| Southern Water | southernwater.co.uk/our-drought-plan | 2019 | 2-9-2020 |
| South Staffs Water | stwater.co.uk/drought-plan | 2019 | 2-9-2020 |
| Sutton and East Surrey Water | seswater.co.uk/publication-drought | 2019 | 2-9-2020 |
| Thames Water | thameswater.co.uk/drought-plan | 2017 | 2-9-2020 |
| United Utilities | unitedutilities.com/drought-plan | 2018 | 2-9-2020 |
| Wessex Water | wessexwater.co.uk/drought-plan | 2018 | 2-9-2020 |
| Yorkshire Water | yorkshirewater.com/resources | 2019 | 2-9-2020 |




## A3 Main water users in England

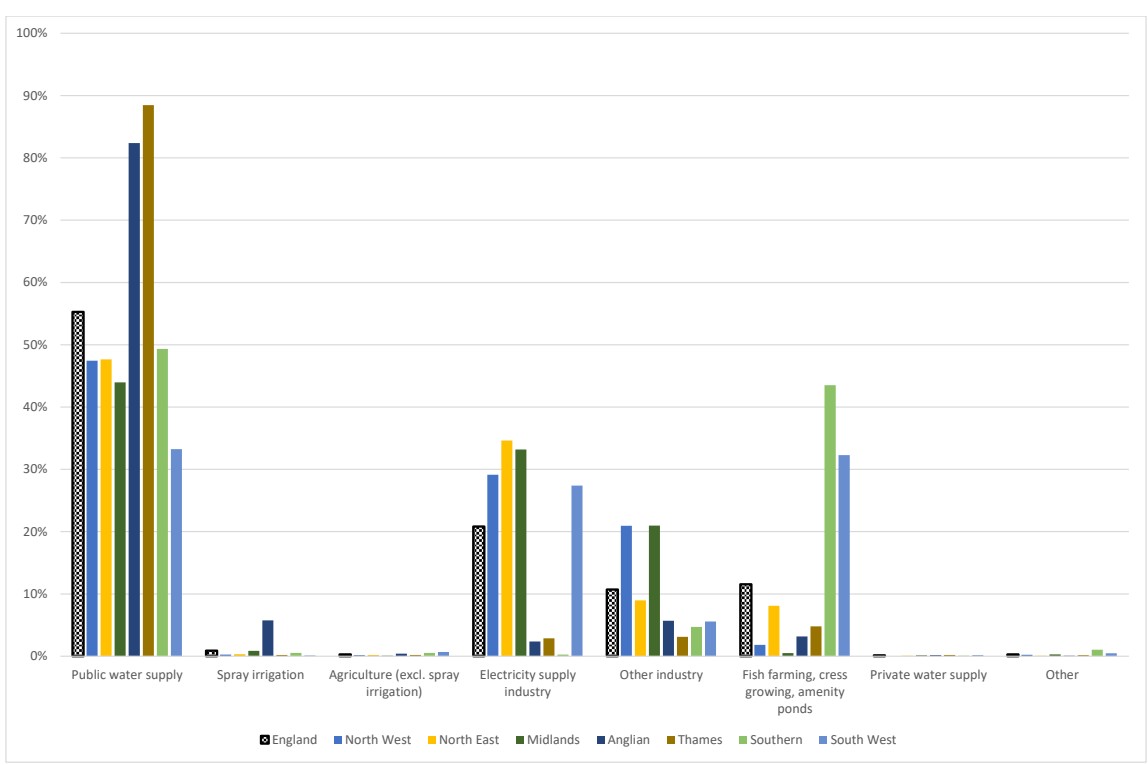

**Figure A1.** Regionally-averaged water users in England (dotted black and white bar) by allocated surface water and groundwater licences (data from 2000-2015; Environment Agency). Regional water use is shown in coloured bars. Data can be found on: https://www.gov.uk/government/statistical-data-sets/env15-water-abstraction-tables (Last accessed on 2-09-2020)





## A4 Inter-annual variation of soil moisture balance in lumped parameter model



**Figure A2.** Inter-annual variation of the soil moisture balance in the socio-hydrological model. The five panels show long-term time series of precipitation actual evapotranspiration, soil moisture, runoff, and groundwater recharge (all in mm). The first 5 years are part of the spin-off period, the remainder (1985-2017) are used in the analysis.





## A5    Natural and human-influenced groundwater storage dynamics (1985-2017)

**Figure A3.** Natural (in black) and human-influenced (in red) conditions of groundwater storage levels in time (1985-2017). The three panels show the high, medium, and low groundwater storage systems. Note that y-axis are different due to the large variation in groundwater storage for each system.

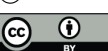
**A6 Surface water storage with combined scenario in the high groundwater storage system and low storage system.**

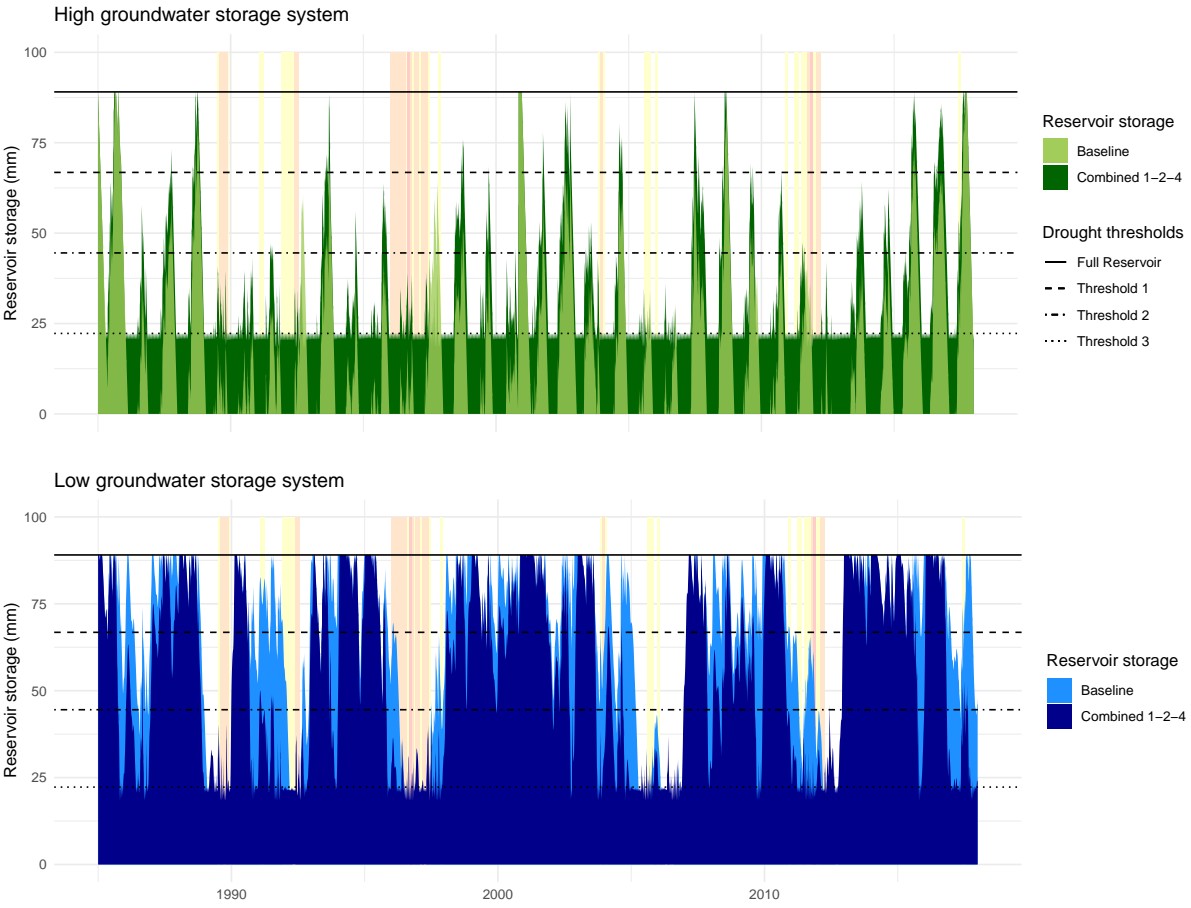

**Figure A4.** Surface reservoir storage in baseline scenario (no drought measures applied) for high groundwater storage catchment (first panel, in light green) and low groundwater storage catchment (second panel, in light blue). Darker green and blue colours indicate the difference in surface water storage as the reservoir is fuller/emptier with the combined scenario (1-2-4; including hands off flow). Coloured surfaces indicate below-normal periods in precipitation (measured in SPI) following Figure 2. Drought thresholds for the surface water reservoir follow the documented range for trigger levels (see Table 1 and Table 3).




## A7 Baseline conditions for groundwater storage under a range of storage-outflow parameters

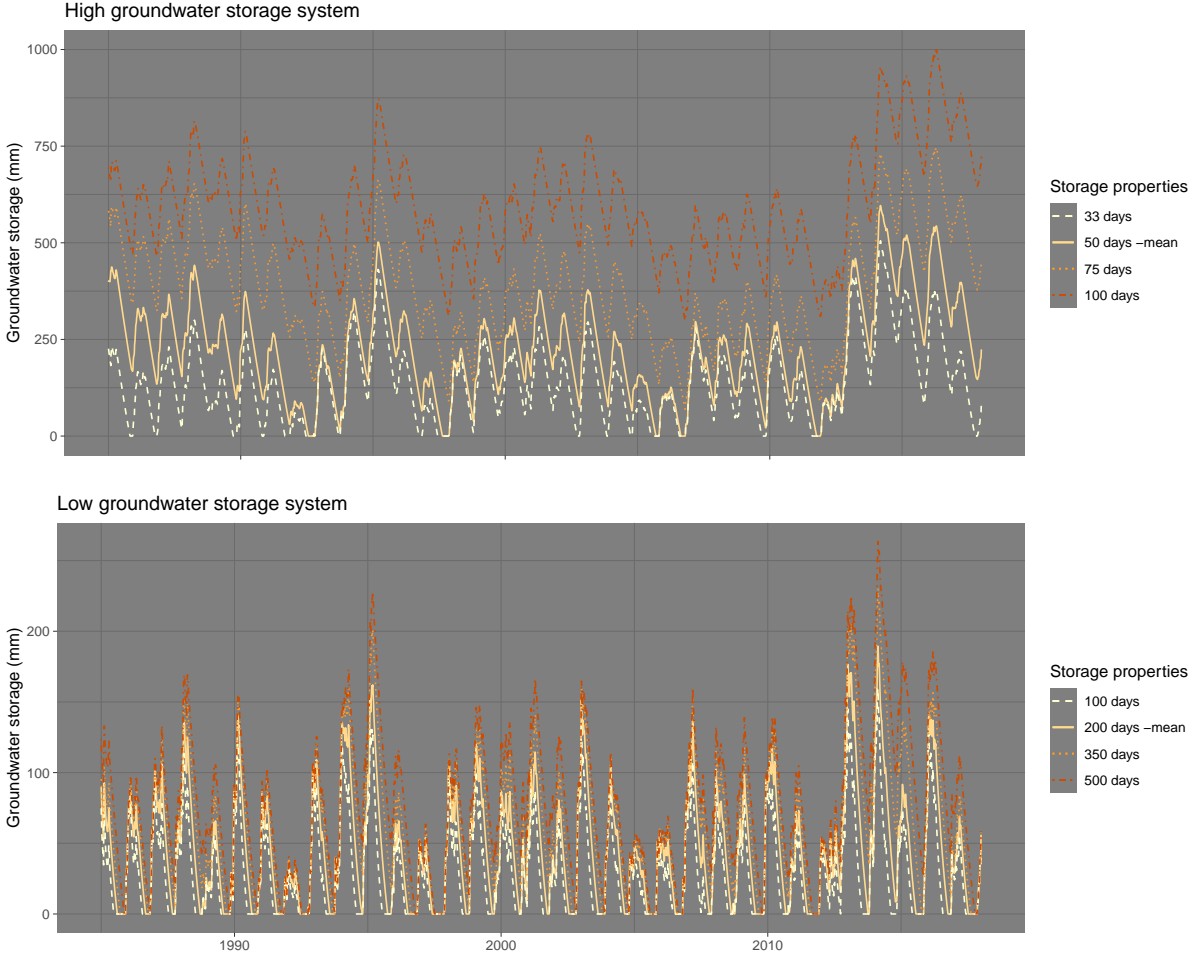

**Figure A5.** Baseline conditions for groundwater storage modelled using different groundwater storage-outflow parameters, as given in Table 2. The first and second panel represent the high and low groundwater storage system.





## A8 Groundwater drought duration and severity for baseline and combined scenarios applying a range of groundwater storage-outflow parameters

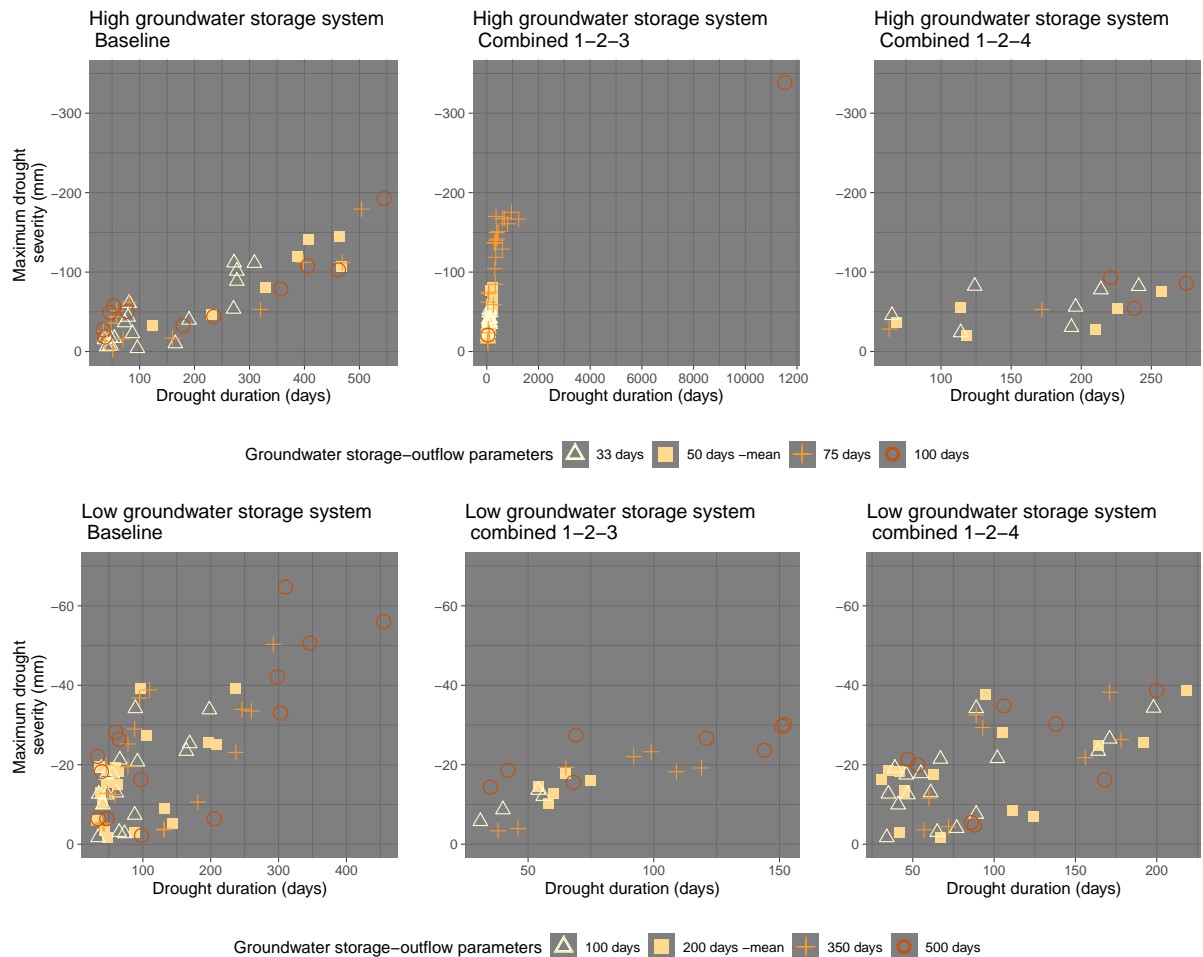

**Figure A6.** Groundwater drought duration and severity for baseline conditions and two combined scenarios (1-2-3 and 1-2-4) in the two groundwater storage systems. The range of groundwater storage-outflow parameters can be found in Table 2.


## A9 Groundwater drought duration and severity for baseline and combined scenarios applying an increase (93%)
and decrease (83.5%) in overall water allocation.

**Figure A7.** Groundwater drought duration and severity for baseline conditions and two combined scenarios (1-2-3 and 1-2-4) in the two groundwater storage systems. These tests are part of the sensitivity analysis for which the proportional water allocation was increased and decreased with 5%.

*Author contributions.* DW has designed and conducted the research in collaboration with MG and BH supervised by AVL, JB and DH. DW has written the manuscript with input from all co-authors. The final version has been approved by all co-authors.



*Competing interests.* Authors have declared no competing interests.

*Acknowledgements.* This paper has been initiated and developed as part of the IAHS Panta Rhei 'Drought in the Anthropocene' working
group. We also acknowledge helpful discussions with Kerstin Stahl, Chris Jackson and Natalie Kieboom. DW thanks the support of CENTA
NERC studentship (NE/lL002493/1), BGS (GA/16S/023) and NERC COVID-19 PhD extension to complete this work. This work also
contributes to the objectives of the NERC-funded 'Groundwater Drought Initiative' (NE/R004994/1). JPB publishes with the permission of
the Executive Director, British Geological Survey (NERC/UKRI).





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
