# Peer review of "Demonstrating the impact of integrated drought policies on hydrological droughts"

_Natural Hazards and Earth System Sciences, 2021_

## Author Comment (AC2)

**General response to Reviewer 1**

We thank the Reviewer for their careful reading and constructive comments to improve the clarity of the submitted manuscript. We will address each comment in detail in the rebuttal, but we have addressed the first five major comments in this Author's response to clarify important aspects of the paper and show how we will include the provided suggestions in the revised
5  manuscript.

**Title and Introduction:**
We agree with the Reviewer that the title could be stronger once reflecting the findings and implications, rather than the impact of drought policies. We will re-evaluate our choice of title and consider rephrasing.
10  Regarding the introduction, we will rephrase the scope to clarify the reasoning behind the focus on hydrological droughts, including both baseflow and groundwater droughts. Because of the specific focus on base flow and groundwater, we considered different hydrogeological settings that are associated with different types of drought characteristics.

**Comment 1: 'High, medium, low groundwater storage systems (L70, L143) are crucial definitions to understand the**
15  **analysis. I wonder if the authors described/used here large, medium and small (or shallow) groundwater storage sys-**
**tems, i.e., characterizing the ability of the system to store more (large) or less (small) water in the subsurface. High/low**
**is rather confusing here as high/low is often used for high or low permeability of the aquifer, i.e., the degree of infil-**
**tration of water into the aquifer. Has a high groundwater system (also) a large storage? This should be clarified (or**
**changed) and a distinct definition of the three systems is needed in a prominent way in the manuscript. Wording should**
20  **be revised also to other sections in the manuscript, e.g., "companies with access to principal aquifers might depend**
**more on groundwater compared to companies with access to shallow, less productive aquifers" (L83-84). '**

We acknowledge the confusion caused by the naming of the three types of groundwater storage systems. We thank Reviewer 1 for highlighting this and we will rename these groundwater systems as suggested. The reasoning behind the high, medium,
25  low naming of groundwater systems is indeed characterising the overall availability of groundwater storage given the modelled groundwater storage-outflow equations. We will include a detailed description in the suggested 'virtual catchment' section to define the large, medium and small groundwater storage systems.

**Comment 2: 'Furthermore, I found the only linkage between hydrogeology and groundwater systems in L139-149. Is**
30  **the linkage only for some kind of justification for different GW model boxes in HBV or was the aim really to quantify**
**the impact of integrated drought policies on hydrological droughts in different hydrogeological settings? For me it is**
**not clear why the hydrogeological features of the virtual catchment (karstic, porous and fractured) are linked 1:1 to**
**high, medium and low groundwater systems? I guess this could be clarified, however, the hydrogeological context could**
**be better integrated in the study. To be honest, my first concern reading the manuscript was on the added value of the**
35  **hydrogeological settings (i.e., karstic, porous, fractured). See also point (4) below.'**

We agree with Reviewer 1 that this should be highlighted earlier, i.e. in the revised introduction. The reason for representing different aquifer types is driven by 1) the aquifer-dependent delay in groundwater storage-outflow (L23-28), 2) the increased dependency on groundwater during droughts (L28-36) and 3) the absent groundwater component in recent drought policy
40  modelling (L52-60). We will summarise these arguments in the last paragraph of the introduction to emphasise the need for different groundwater systems in this study.
Previous hydrological drought modelling by Van Lanen et al. (2013) applied a modified the standard HBV model to simulate hydrological droughts globally and changed the response time (in days) to represent different groundwater systems, finding that the responsiveness of groundwater systems has a large impact on drought characteristics. Stoelzle et al. (2015) extended
45  the representation of different aquifer structures for a lumped model approach by identifying superior groundwater simulation structures. They tested and recommended a range of alternative model structures for five aquifer types (see our response to Comment 4 for details). Out of these five aquifer types, we modelled three aiming to show the impact of drought management strategies for different groundwater systems.

The three hydrogeological settings (karstic, porous and fractured aquifers) were selected to broadly representative for catchments in England and therefore excluded the 'mixed' and 'combined' aquifer types of Stoelzle et al. (2015). The first hydrogeological setting is generally associated with a large groundwater storage availability and non-linear drainage as found in karstic aquifers (Bloomfield and Marchant, 2013; Hartmann et al., 2014). Medium groundwater storage availability was found when modelling the porous aquifer type with slow drainage and possible leakage (Shepley et al., 2008; Allen et al., 1997). The last setting represented smaller groundwater storage availability with short response times for fractured or weathered aquifers (Allen et al., 1997). In modelling these three types of aquifers, none of the groundwater storage potential was constrained and different baseflow groundwater storage resulted from the different groundwater storage-outflow equations. In the revised manuscript, we will rephrase the introduction of these three groundwater systems and highlight the link to these different hydrogeological settings.

**Comment 3: 'I suggest to have a separated section "virtual catchment" where the modelling approach is explained (in single steps) using HBV (with a specific model structure) a set of average forcing data for England. Do I understand it correctly that no calibration was done in HBV as fixed parameter values were derived from literature etc. and there is no observed runoff? This should be mentioned more clearly! Consistent terms would be beneficial (at the moment idealized, simplified and virtual catchment is used). Perhaps this could also be done with an extension of Fig.1 showing (a) the HBV model structure (+ extension with three different GW boxes) and forcing data and (b) the sociohydrological model approach next to each other.'**

We appreciate the suggested section with the heading 'virtual catchment', as this will indeed clarify the scope of the paper and associated modelling assumptions related to the three hydrogeological settings. As stated in L144-149, modelled groundwater storage-outflow parameters were based on the tested parameter range by Stoelzle et al. (2015) and mean aquifer characteristics in England (Allen et al., 1997). A wider range of parameters was tested in the sensitivity analysis, which results can indeed be better included in earlier sections, as suggested in Comment 5. We will rephrase this in the Results section and include findings of the sensitivity analysis were relevant.

As noted by Reviewer 1, there is no comparison to observed discharge or groundwater storage in this idealised hydrological system (L64-66). We used the term 'idealised' referring to the simplified hydrological system that can be seen as a stand-alone simplified example used for analysis, hence the term 'virtual catchment'. We will review the use of these terms for consistency. Given the stand-alone example and modelling exercise, there is no validation of the performance of the different aquifer structures in this virtual modelling study that is merely building on from the validated model structure to assess hydrological droughts (Van Lanen et al., 2013).

Lastly, we have revised Fig. 1 to reflect both validated model structure, different groundwater modules and environmental and anthropogenic water demand. All three components have been included in the revised Figure 1 (see below).

**Comment 4: 'What is the advantage to have three different GW model representations for the three different groundwater systems if also variation of the parameterization of the same model structure could do this job? Stoelzle et al. (2015) showed that the FLEX GW structure outperforms the three structures POW, 1LBY and 2PA used in this study and the hydrogeological clustering of catchments is also better/clearer for FLEX than for other structures (Fig. 4 in Stoelzle et al., 2015). Wouldn't it be easier to compare the effects of larger and smaller groundwater systems on different drought policies if the model structures across those different GW systems stay the same?'**

It is a valid point raised by Reviewer 1 that when model validation is possible, all three groundwater systems can be well-simulated using the FLEX GW structure. However, in this virtual catchment modelling we aimed to represent different groundwater systems without the need for validation to find a representative share of fast and slow responding groundwater discharge. Therefore, the FLEX GW structure is not ideal given the dependency on the threshold-controlled storage outflow (h) to determine either primarily fast or slow groundwater discharge response. As an alternative, we applied suggested conceptual model structures to represent different kinds of groundwater storage-outflow characteristics that typically result in different drought characteristics for karstic, porous and fractured aquifer types. The non-linear groundwater discharge release, as observed in karstic aquifers, is best represented by a power law (POW) (Wittenberg, 2003). Slow porous flow could be represented by a

[Figure]

**Figure 1.** Socio-hydrological model consisting of a soil moisture balance driven by precipitation (P in mm d$^{-1}$) and potential evaporation (PET in mm d$^{-1}$), a surface water reservoir storing runoff (Qr mm d$^{-1}$), and a groundwater module that consists of three groundwater system options (large, medium, small groundwater availability) driven by groundwater recharge (Rch in mm d$^{-1}$). Anthropogenic water demand is met by reservoir abstractions (Ares in mm d$^{-1}$ ) and groundwater abstractions (Agw in mm d$^{-1}$), both in striped dark red arrows. Natural water demand is represented by ecological flow requirements (Qeco in mm d$^{-1}$; dotted green arrow) and abstracted as part of the baseflow (Qb in mm d$^{-1}$). Remaining baseflow is routed to the reservoir. Additional water is imported in the model when reservoir or groundwater storage is insufficient (Qimp and GSimp both in mm d$^{-1}$). Drought management scenarios apply to the surface water reservoir, groundwater module, and environmental and anthropogenic water demand (all model components in the thick black box).

number of structures, but given the overall high performance by 1LBY (Figure 5 in Stoelzle et al. (2015)) and the reported slow flow and possibly leakage in English Permo-Triassic sandstone aquifers, we selected the 1LBY. For fractured aquifers, the overall recommendation to apply parallel reservoirs and the overall high performance of 2PA (with groundwater recharge input) were the reasons for selecting 2PA for the last groundwater system.

By selecting these three conceptual model structures, we aimed to represent three different kinds of groundwater storage-outflow release whilst testing a range of representative parameters for English catchments. In the revised manuscript, we will include the reasoning behind the selection of model structures in a virtual catchment.

**Comment 5:** 'I like the section 5.3 model limitations as this discussion is really important to understand the results of the study. Beside the fact that this section could be incorporated into other sections of the manuscript or at least should not be placed at the end of the discussion section (to gain a more positive ending of the paper), I asked myself

**what would have been happened if another forcing than an England's average was used? Are average conditions a good**
**starting point for such drought analysis like here? Additional analysis could shed light on this, however, at least more**
**discussion is needed to evaluate how representative the average approach is for the different regions in England or**
**different water companies.'**

We thank Reviewer 1 for these two suggestions. We will include relevant results of the sensitivity analysis in earlier sections if relevant (also see response to comment 2). Regarding the second suggestion, our intention was to select a representative precipitation record for England. The HadUKP data consists of weighted observations, including extremely wet and dry periods within its record (Alexander and Jones, 2001). However, we will verify if other forcing English data, i.e. a representative location in the CEH-GEAR dataset (Tanguy et al., 2016) changes the overall water balance and outcomes. We will include these additional analyses in the full rebuttal and amend the results in revised manuscript if necessary.

**References**

Alexander, L. and Jones, P.: Updated precipitation series for the UK and discussion of recent extremes, Atmospheric science letters, 1, 142–150, 2001.

Allen, D., Bloomfield, J., and Robinson, V., eds.: The physical properties of major aquifers in England and Wales, British Geological Survey (WD/97/034), 1997.

Bloomfield, J. P. and Marchant, B. P.: Analysis of groundwater drought building on the standardised precipitation index approach, Hydrology and Earth System Sciences, 17, 4769–4787, https://doi.org/10.5194/hess-17-4769-2013, 2013.

Hartmann, A., Goldscheider, N., Wagener, T., Lange, J., and Weiler, M.: Karst water resources in a changing world: Review of hydrological modeling approaches, Reviews of Geophysics, 52, 218–242, https://doi.org/https://doi.org/10.1002/2013RG000443, 2014.

Shepley, M., Pearson, A., Smith, G., and Banton, C.: The impacts of coal mining subsidence on groundwater resources management of the East Midlands Permo-Triassic Sandstone aquifer, England, Quarterly Journal of Engineering Geology and Hydrogeology, 41, 425–438, https://doi.org/10.1144/1470-9236/07-210, 2008.

Stoelzle, M., Weiler, M., Stahl, K., Morhard, A., and Schuetz, T.: Is there a superior conceptual groundwater model structure for baseflow simulation?, Hydrological processes, 29, 1301–1313, 2015.

Tanguy, M., Dixon, H., Prosdocimi, I., Morris, D. G., and Keller, V. D. J.: Gridded estimates of daily and monthly areal rainfall for the United Kingdom (1890-2015), https://doi.org/10.5285/33604ea0-c238-4488-813d-0ad9ab7c51ca, 2016.

Van Lanen, H. A. J., Wanders, N., Tallaksen, L. M., and Van Loon, A. F.: Hydrological drought across the world: impact of climate and physical catchment structure, Hydrology Earth System Sciences, 17, 1715–1732, https://doi.org/10.5194/hess-17-1715-2013, 2013.

Wittenberg, H.: Effects of season and man-made changes on baseflow and flow recession: case studies, Hydrological Processes, 17, 2113–2123, https://doi.org/https://doi.org/10.1002/hyp.1324, 2003.

---

## Author Comment (AC3)

**General response to Reviewer 2**

We thank the Reviewer for the positive feedback and minor comments. It is encouraging to read that the current synthetic model setup is reviewed as a good representative for a UK water resource system. However, we do acknowledge that some aspects of the water resources system are missing (see our response to comment 4) or at least simplified (see our response to comment 5). We will review the discussion to make sure recommendations are supported by the sensitivity analysis. Overall, we have provided a first author response to the main comments (1-6) here and will address comments (7-12) directly in the point-wise rebuttal later.

**Comment 1: As I mention, modelling water resources with groundwater in a joined up way is surprisingly rare - the authors may wish to include a slightly expanded review (even just a paragraph in the intro) on this subject to help justify their reasonably simplistic hydrological representation.**

We thank Reviewer 2 for noting the value for joining water resources modelling and groundwater and appreciate their suggestion of a (short) paragraph in the introduction to emphasise this.

**Comment 2: Can water demand be written as an equation, just to make it easier to follow**

Agreed. This is a good point, as adding an equation for the total water demand is more consistent with the overall modelling approach and will improve the manuscript. The suggested demand equation might also clarify the term 'headroom' as Reviewer 1 mentioned (Comment 7).

**Comment 3: L116/133 - Perhaps either move some of the text from S3.2 here, or at least reference that this is described in more detail in the Data section. On first reading it appeared that this line was essentially the only description of water demand in the model!**

We agree with Reviewer 2 that there is some information missing in the Model Structure section. We will add some explanation regarding the water management plans (now in the Data section) and refer to the overview Table A1 in the supplementary material.

**Comment 4: I was under the impression that ecological flows are typically met by reservoir releases, rather than groundwater pumping (though I suppose this is highly regional). No need to rerun the model, but might be interesting for UK readers.**

This is an interesting point by Reviewer 2, as maintaining ecological flow requirements can refer to either reservoir release or restricting groundwater abstractions supporting surface water and wetlands, or possibly a combination of both options depending on the position of reservoir in the catchment and connectivity of the stream and aquifer (Environment Agency, 2019). Based on the thirteen drought management plans that source water from both surface water and groundwater, we found that groundwater abstractions are restricted when baseflow falls below a certain ecological minimum flow threshold. During severe droughts, drinking water companies can apply for a drought order to sustain groundwater abstractions potentially lowering ecological flows. This is why we modelled the 'hands off flow' scenario applying this restricted use of groundwater. We will clarify this groundwater-related focus on maintaining the ecological flow in the scenario description in section 3.2.
When simulating a specific catchment setting or alternative modelling approach, the 'hands off flow' also could include a fixed reservoir release or a combination of restricted groundwater use and reservoir release depending on the relevant catchment setting. We will mention this alternative approach when suggesting alternative modelling assumptions in section 5.3 (model limitations in L431-437).

**Comment 5: The groundwater and reservoir levels in this model are often 0 (Fig 2). This is no problem since the case study is synthetic, but the authors should note in the text that UK groundwater/reservoir systems are not this stressed (even if effort has been taken to parameterise the model in a sensible and nationally reflective manner) - perhaps expand a little in Section 5.3 (water companies might be alarmed if you give the impression that this is portrayed as a 'nationally average' model)**

We agree with the Reviewer that particularly the low groundwater storage system is quite stressed *in the baseline scenario*

given the synthetic model settings. This is likely a combination of a fast responding aquifer modelled in a lumped model and
considerable pressure on the water system *without any management interventions* in place. In the lumped model, the available groundwater storage excludes deeper groundwater sections or lateral groundwater flow (L170-171 in old manuscript) that results in zero storage when the storage capacity of an aquifer is low and fast responding and groundwater demand is high. Even though the pressure on water resources is based on actual water resource management plans, the use of water resources is static in the baseline scenarios based on the average of thirteen drinking water companies and fixed for the 37-year modelling period. Scenarios including conditional/flexible of surface water and groundwater might be a better representation of water management practises in England considering the low flow alleviation scheme in place (Environment Agency, 2016; Howarth, 2018) and the relatively flexible combination of both surface water and groundwater in larger water management regions (Shepley et al., 2009; Fowler et al., 2007; Thorne et al., 2003). Therefore, modelled groundwater and reservoir levels in scenarios that include either conjunctive use or 'hands off flow' are likely to be a closer representation of the status of water resources in England. As suggested, we will expand on these aspects in section 5.3 and remove statements suggesting that the unmanaged (baseline) condition reflect the status of water resources in England.

**Comment 6: Can the model/modelling setup be made publicly available so that results can be reproduced - it seems that the models/data are all openly available so I don't see why not?**

We agree with the Reviewer that this possible given the open data and use of published models. However, more work is still required to meet open science and coding standards, which are difficult to meet withing the time frame of the current employment contract.

**References**

Environment Agency: Managing water abstraction, 2016. https://www.gov.uk/government/publications/managing-water-abstraction/managing-water-abstraction

Environment Agency: River Basin Management Plan - water levels and flows challenge, 2019. https://consult.environment-agency.gov.uk/++preview++/environment-and-business/challenges-and-choices/user_uploads/the-economics-of-managing-water-rbmp-2021.pdf

Fowler, H., Kilsby, C., and Stunell, J.: Modelling the impacts of projected future climate change on water resources in north-west England, Hydrology Earth System Sciences, 11, 1115–1126, 2007.

Howarth, W.: Going with the flow: Integrated Water Resources Management, the EU Water Framework Directive and ecological flows, Legal Studies, 38, 298–319, https://doi.org/10.1017/lst.2017.13, 2018.

Shepley, M., Streetly, M., Voyce, K., and Bamford, F.: Management of stream compensation for a large conjunctive use scheme, Shropshire, UK, Water and Environment Journal, 23, 263–271, https://doi.org/10.1111/j.1747-6593.2008.00158.x, 2009.

Thorne, J. M., Savic, D. A., and Weston, A.: Optimised Conjunctive Control Rules for a System of Water Supply Sources: Roadford Reservoir System (U.K.), Water Resources Management, 17, 183–196, https://doi.org/10.1023/A:1024157210054, 2003.

---

## Author Response (AR1)

**Editor**

**Dear authors,**

**I'm very glad that your submitted manuscript developed that well. I highly appreciate both of the very detailed and helpful reviews. Many thanks for your time and your will to further improve this work. I fully agree with the points raised by the referees. Since you already explicitly replied to the reviews I please you to carefully follow the referee's suggestions as proposed and I'm looking forward to see your manuscript after minor revisions. All the best, Veit.**

Dear Veit,

Thanks for inviting revised manuscript after minor revisions. We appreciated both reviews that have helped to improve the manuscript. As you recommended, we have followed the suggestions closely. Please find the point-wise rebuttal below (reviewer comments numbered and in **bold**) and the track-changes document and revised manuscript via the NHESS portal.

On behalf of the co-authors,

Doris Wendt

**Reviewer 1**

**This paper by Wendt et al. investigates the effect of different drought policies on hydrological drought characteristics and is a relevant contribution for the community and water and drought management. The study uses a virtual catchment and embeds different hydrogeological settings in a model experiment. Different drought strategies are then implemented as different scenarios to simulate streamflow and groundwater storages under different conditions. The scenarios are well-chosen and help to evaluate different challenges in future drought management. I recommend publication after moderate revisions.**

We thank Reviewer 1 for their careful reading and constructive comments to improve the clarity and consistency in the manuscript. We think that these comments have increased the overall quality of the work. Particularly the suggested additional heading, renaming of groundwater systems and suggestions for the figures were very constructive. These improvements and other changes to the submitted manuscript can be found below in the point-wise rebuttal (comments are numbered R1C..).

**R1C2: Title: For me the paper title would be more informative/stronger if it states what it is found rather than what is done in the study. I suggest to go into the direction of "Integrated drought policies on hydrological drought led to ..." However, the readers might be more interested in the "implications" of the study than in a "demonstration of the impact".**

We thank Reviewer 1 for this suggestion and have revised the title to represent the main findings of the modelling study. The new title is 'Evaluating integrated water management strategies to inform hydrological drought mitigation'. With the revised title we intend to direct readers directly to the main findings without going into technical (modelling) details that might meet a broader NHESS audience.

**R1C2: Introduction: I liked the condensed format of the introduction and a lot of recent publications are embedded there. However, it remains unclear for me whether the focus of this study is on hydrological droughts, groundwater (L35-36) or on both, or human modified droughts (L31) – this should be clarified (although it gets clearer later on in the manuscript). "Hydrogeological conditions" are mentioned the first time in L64 in the introduction (in the Study Aims paragraph). How is it justified to consider different hydrogeological settings? This might be a logical approach for a drought researcher, but should be referenced and/or better introduced for a broader readership. There is also a mixture of objectives and method description (L63-71) in the introduction. Clear, tailored objectives or research questions could help to explain what exactly the science is in this paper? The key elements of drought policies (L39-41) are given but no further used in the introduction. Why is the list of those six elements important?**

We thank the reviewer for their constructive comments and have revised the introduction to emphasise the focus of the paper. We refer now first to baseflow and groundwater droughts in the abstract (L7-8) and define the term hydrological droughts within the research aim (L72-75, see below).

In response to the application of different hydrogeological conditions, we have introduced a short paragraph in the introduction to emphasise the complexity of groundwater availability and need for evaluation of both surface water and groundwater in drought policies.

> L59-69:Given the increasing dependency on groundwater storage during meteorological droughts (Aeschbach-Hertig and Gleeson, 2012; Taylor et al., 2013; Cuthbert et al., 2019), drought policy modelling should include both surface water and groundwater, to reflect the additional complexity of possibly contrasting groundwater storage availability within or between water management regions. In natural systems, temporal variation in groundwater storage and aquifer-dependant delay in groundwater storage and baseflow results in contrasting baseflow and groundwater drought characteristics (Peters et al., 2006; Van Lanen et al., 2013; Bloomfield and Marchant, 2013). These contrasting hydrological drought characteristics change when impacted by managed groundwater use (Tijdeman et al., 2018; Wendt et al., 2020) and overall drought resilience reduces when groundwater use exceeds sustainable limits (Custodio, 2002; Custodio et al., 2019). On the other hand, targeted management strategies can also ease pressure on groundwater systems (Klaar et al., 2014; White et al., 2019) and encourage integrated water use aiming to increase drought resilience (Huggins et al., 2018; Scanlon et al., 2016; Jakeman et al., 2016), highlighting their potential within drought policies.

In addition to this, we have also rephrased the last paragraph in the introduction, elaborating on the aim of the research and how the scenarios contribute to this. We appreciate the constructive comment, as these revisions lead to broader readership compared to the submitted manuscript.

> L71-79: These conditions refer to the availability of groundwater storage in a (virtual) catchment that is modelled for groundwater systems with overall large, medium and small groundwater availability. Hydrological droughts represent both baseflow and groundwater that might be either human-modified or human-induced droughts (Van Loon et al., 2016), as a consequence of water management (baseline) or drought management strategies, which are introduced either in separate or combined drought management strategies in a virtual socio-hydrological model. This socio-hydrological model represents an idealised virtual hydrological system that includes a surface water reservoir, a groundwater module with either large, medium or small groundwater storage availability and an option to import surface water to meet either anthropogenic or environmental water demand.

**R1C3: High, medium, low groundwater storage systems (L70, L143) are crucial definitions to understand the analysis. I wonder if the authors described/used here large, medium and small (or shallow) groundwater storage systems, i.e., characterizing the ability of the system to store more (large) or less (small) water in the subsurface. High/low is rather confusing here as high/low is often used for high or low permeability of the aquifer, i.e., the degree of infiltration of water into the aquifer. Has a high groundwater system (also) a large storage? This should be clarified (or changed) and a distinct definition of the three systems is needed in a prominent way in the manuscript. Wording should be revised also to other sections in the manuscript, e.g., "companies with access to principal aquifers might depend more on groundwater compared to companies with access to shallow, less productive aquifers" (L83-84).**

We thank the reviewer for highlighting the potential confusing in naming of the three groundwater storage systems. The reasoning behind the high, medium, low naming of groundwater systems is indeed characterising the overall *availability* of groundwater storage given the modelled groundwater storage-outflow equations. We have highlighted this in the last two paragraphs in the introduction (see R1C2) and have revised the naming into large-medium-small groundwater storage availability. We have also rephrased lines L83-84 (now L90-92) for consistency.

> L90-92: For example, in regions with large groundwater storage availability water supply might rely more on groundwater compared to regions with smaller groundwater storage availability. In England this regional variability is reflected in the share of either surface water or groundwater for the thirteen drinking water companies (Table A1).

**R1C4: Furthermore, I found the only linkage between hydrogeology and groundwater systems in L139-149. Is the linkage only for some kind of justification for different GW model boxes in HBV or was the aim really to quantify the**

impact of integrated drought policies on hydrological droughts in different hydrogeological settings? For me it is not clear why the hydrogeological features of the virtual catchment (karstic, porous and fractured) are linked 1:1 to high, medium and low groundwater systems? I guess this could be clarified, however, the hydrogeological context could be better integrated in the study. To be honest, my first concern reading the manuscript was on the added value of the hydrogeological settings (i.e., karstic, porous, fractured). See also point (4) below.'

We agree with Reviewer 1 that this aspect needed clarification and an earlier mention in the manuscript. We have clarified this by including an additional paragraph in the introduction and additional explanation in the last paragraph of the introduction (L59:69, see also R1C2 and R1C3).

Regarding the modified HBV structure, we revised the text in model structure section (see R1C5 for L126-127) to emphasise the starting point of the socio-hydrological model, i.e. the hydrological drought modelling of Van Lanen et al. (2013).

In the revised manuscript, the direct link between the three parallel groundwater systems is provided (L169-175) describing how we extended this by applying three different groundwater options (in parallel) in the groundwater module based on the work of Stoelzle et al. (2015). This extension advances current drought policy modelling, as the different parallel groundwater storage options allow a comparison of the impact of drought management strategies for a range of large, medium and small groundwater availability. The first hydrogeological setting is generally associated with a large groundwater storage availability and non-linear drainage as found in karstic aquifers (Bloomfield and Marchant, 2013; Hartmann et al., 2014). Medium groundwater storage availability was found when modelling the porous aquifer type with slow drainage and possible leakage (Shepley et al., 2008; Allen et al., 1997). The last setting represented smaller groundwater storage availability with short response times for fractured or weathered aquifers (Allen et al., 1997). In modelling these three types of aquifers, none of the groundwater storage potential was constrained. In the revised manuscript, we will rephrase the introduction of these three groundwater systems and highlight the link to these different hydrogeological settings.

*L169-175: The groundwater module has three different parallel options for groundwater storage availability, representing different hydrogeological conditions. The first option is named 'large groundwater storage system' referring to an overall large groundwater availability, as typically found in karstic groundwater systems (Stoelzle et al., 2015; Hartmann et al., 2014). The second option in the groundwater module is the 'medium groundwater storage system' referring to medium groundwater availability, as can be found in porous aquifers (Allen et al., 1997; Bloomfield and Marchant, 2013; Stoelzle et al., 2015). The last option is 'small groundwater storage system' referring to small groundwater availability typically found in shallow or weathered fractured aquifers (Allen et al., 1997; Stoelzle et al., 2015).*

**R1C5: I suggest to have a separated section "virtual catchment" where the modelling approach is explained (in single steps) using HBV (with a specific model structure) a set of average forcing data for England. Do I understand it correctly that no calibration was done in HBV as fixed parameter values were derived from literature etc. and there is no observed runoff? This should be mentioned more clearly! Consistent terms would be beneficial (at the moment idealized, simplified and virtual catchment is used). Perhaps this could also be done with an extension of Fig.1 showing (a) the HBV model structure (+ extension with three different GW boxes) and forcing data and (b) the sociohydrological model approach next to each other.'**

We appreciate the suggested section and have added a short section 'virtual socio-hydrological model'. This new sub-header describes the modelling setup, representative forcing data and management setting and flow of forcing data though the model (modelling steps) in sequence (125-138). This separate section emphasises that the results are stand-alone and representing a virtual setting. Reservoir levels, baseflow and groundwater levels are thus not calibrated.

*L125-138: The virtual socio-hydrological follows a standard conceptual water balance model with additional water demand components (Figure 1). The water balance model was based on the previously described lumped hydrological model of Van Lanen et al. 2013, who modified the standard HBV model structure (Bergström, 1976) to model hydrological droughts globally. We extended this hydrological drought model with three different groundwater storage options in the groundwater module and introduced surface water demand and/or groundwater demand that could be altered following a drought management plan. Forcing data in the virtual model was selected to*

*be representative for the case study (England) and management settings and scenarios were likewise based on a*
*range of water management and drought management plans converted to relative setting to be applied in the vir-*
*tual socio-hydrological model. In sum, the virtual socio-hydrological model is thus driven by English climate data*
*that drives the daily soil moisture balance, generating runoff an groundwater recharge. Runoff is directly routed*
*to the surface water reservoir. Groundwater recharge is either stored or discharged depending on the groundwa-*
*ter storage option in the groundwater module. Water demand is met using a proportion of stored surface water*
*and/or groundwater that can be imported externally in the model when storage is depleted. Drought management*
*scenarios can alter the proportion of water demand and source of water supply that has an impact on hydrological*
*droughts and water resource availability.*

We used the term 'idealised' referring to the simplified hydrological system that can be seen as a stand-alone simplified exam-
ple used for analysis, hence the term 'virtual catchment'. We have revised the text for consistency in the last paragraph in the
introduction (see R1C2).

**R1C5- follow up suggestion - The revised Fig. 1 is a great move forward. I suggest to replace the reservoir design of
"Groundwater module" with a normal box (as it is a representation). Then, the three options below are a great expla-
nation (and they are actually storage systems in the model!), please add "groundwater aquifer" to "Large", "medium"
and "small" and perhaps something like power-law, by-pass and two-parallel to the storage boxes (and the hydrogeo-
logical representations). The two arrows in "medium" and "small" can be confused, is the by-pass flowing through the
storage, is the outflow of the first parallel storage added to the second?**
In response to the first and follow-up suggestions, we have revised the conceptual diagram in Figure 1 to show the validated
model structure, different groundwater storage options in the groundwater module, and the environmental and anthropogenic
water demand, see the revised Figure 1. In the Figure caption, the names of the groundwater systems are linked to the power
law, by-pass, and parallel storage-outflow equations and referred to a detailed description of the hydrogeological representa-
tions in the manuscript. In response to the last suggestion regarding the arrows in the medium and small groundwater storage
systems, we have changed the layout from the first posted revised Figure 1 online to avoid further confusion. In the detailed
description in L185-188, we link the indicated by-pass storage-outflow (10%) to the additional arrow that bypasses the ground-
water storage. The two parallel storage reservoir in the small groundwater storage system are modelled in parallel, for which
groundwater recharge and water demand is equally divided (L190-193 in revised manuscript).

[Figure]

**Figure 1.** Virtual socio-hydrological model consisting of a soil moisture balance driven by precipitation (P in mm d$^{-1}$) and potential evaporation (PET in mm d$^{-1}$), a surface water reservoir storing runoff (Qr mm d$^{-1}$), and a groundwater module that consists of three groundwater system options (large, medium, small groundwater availability) driven by groundwater recharge (Rch in mm d$^{-1}$). These three groundwater systems represent large, medium and small groundwater availability, modelled by a power law, by-pass and two parallel reservoir storages, respectively (see 3.2 for details). Anthropogenic water demand is met by reservoir abstractions (Ares in mm d$^{-1}$) and groundwater abstractions (Agw in mm d$^{-1}$), both in striped dark red arrows. Natural water demand is represented by ecological flow requirements (Qeco in mm d$^{-1}$; dotted green arrow) and abstracted as part of the baseflow (Qb in mm d$^{-1}$). Remaining baseflow is routed to the reservoir. Additional water is imported in the model when reservoir or groundwater storage is insufficient (Qimp and GSimp both in mm d$^{-1}$). Drought management scenarios apply to the surface water reservoir, groundwater module, and environmental and anthropogenic water demand (all model components in the thick black box).

**R1C6: What is the advantage to have three different GW model representations for the three different groundwater systems if also variation of the parameterization of the same model structure could do this job? Stoelzle et al. (2015) showed that the FLEX GW structure outperforms the three structures POW, 1LBY and 2PA used in this study and the**
170    **hydrogeological clustering of catchments is also better/clearer for FLEX than for other structures (Fig. 4 in Stoelzle et al., 2015). Wouldn't it be easier to compare the effects of larger and smaller groundwater systems on different drought policies if the model structures across those different GW systems stay the same?**

It is a valid point raised by Reviewer 1 that when model validation is possible, all three groundwater systems can be well-simulated using the FLEX GW structure. However, we use a virtual catchment to represent different groundwater systems. The
175    advantage of using three different GW model representations is that we can represent different groundwater systems without the need for validation to find a representative share of fast and slow responding groundwater discharge. Therefore, the FLEX GW structure is not ideal given the dependency on the threshold-controlled storage outflow (h) to determine either primarily fast or slow groundwater discharge response.

By selecting three conceptual model structures, we aimed to represent three different kinds of groundwater storage-outflow release whilst testing a range of representative parameters for English catchments. The applied GW model representations structures for karstic, porous and fractured aquifers were selected based on the recommendations of Stoelzle et al. (2015) and related literature. Regarding the high groundwater storage availablity and non-linear groundwater discharge release, as typical in a karstic aquifer, we applied the recommended power law (POW) GW structure (Wittenberg, 2003). Slow porous flow could be represented by a number of structures, but given the overall high performance by 1LBY (Figure 5 in Stoelzle et al. (2015)) and corresponding slow porous flow with possible leakage in the English Permo-Triassic sandstone aquifer, we selected the 1LBY (Shepley et al., 2008; Allen et al., 1997). For fractured aquifers, the overall recommendation to apply parallel reservoirs and the overall high performance of 2PA (with groundwater recharge input) were the reasons for selecting 2PA for the last groundwater system. In the revised manuscript, we have included the reasoning behind the selection of model structures in L175-193.

*L175-193: These three parallel options are modelled using different model structures corresponding to a typical karstic, porous and fractured groundwater-discharge release (Stoelzle et al., 2015). Modelled storage-discharge parameters (s in $d^{-1}$ in Table 2) are based on average characteristics found in English karstic, porous, and fractured aquifers (Allen et al., 1997) and tested parameters by Stoelzle et al. (2015). These two ranges of relevant storage-discharge parameters resulted in a mean s parameter for the main result section with a large range tested in the sensitivity analysis.*

*The large groundwater storage system was modelled by a non-linear power law (Equation 5) representing the non-linear groundwater release in karstic aquifers (Wittenberg, 2003; Stoelzle et al., 2015). The non-linearity of discharge release was taken as 0.5 (B in Equation 5) allowing some turbulent flow that is typical for unconfined karstic aquifers (Wittenberg, 2003).*

*The medium groundwater storage system is represented by a linear storage reservoir with additional by-pass component (D; Equation 6) that corresponds to the typical slow porous flow with possible leakage in English Permo-Triassic sandstone aquifers (Shepley et al., 2008; Allen et al., 1997). Possible leakage of groundwater recharge represents 10% based on the tested range (0.07-0.12) by Stoelzle et al. (2015).*

*The small groundwater storage system is represented by two parallel linear storage reservoirs (Equation 7), referring to weathered, fractured aquifers with variable storage-outflow release (Stoelzle et al., 2015; Allen et al., 1997). When applying this option in the groundwater module, total groundwater storage is a sum of both parallel storage reservoirs with different s parameter values, for which recharge and water demand is equally divided.*

**R1C7: I like the section 5.3 model limitations as this discussion is really important to understand the results of the study. Beside the fact that this section could be incorporated into other sections of the manuscript or at least should not be placed at the end of the discussion section (to gain a more positive ending of the paper), I asked myself what would have been happened if another forcing than an England's average was used? Are average conditions a good starting point for such drought analysis like here? Additional analysis could shed light on this, however, at least more discussion is needed to evaluate how representative the average approach is for the different regions in England or different water companies.**

We thank Reviewer 1 for these two suggestions and we have included relevant results of the sensitivity analysis in earlier sections where in section 4.3 (see track-changes version of the revised manuscript).

Regarding the second suggestion, our intention was to select a representative precipitation record for England. The HadUKP data consists of weighted observations, including extremely wet and dry periods within its record (Alexander and Jones, 2001). However, we have verified if another representative location for daily precipitation, i.e. the centroid location used for potential evaporation would result in different results. When using the centroid location to extract a cell of the CEH-GEAR dataset (Tanguy et al., 2016) the overall water balance is 7mm compared to 18mm for 37 years when using the HadUKP data (L254-255). Given the small overall difference to the water balance, we are confident that the HadUKP data is suitable forcing data for this purpose. However, when applying the virtual to specific catchment/regions, it is recommended to re-run the model with relevant climate data (L470-472).

**R1C8: Regarding the publication in NHESS my first impression was that a more hydrological journal such as HESS would be a better choice for this manuscript. Terms like risk, hazard, vulnerability are not or seldom embedded in this manuscript. But as the paper is planned to be published in the SI: "Drought vulnerability, risk, and impact assessments: bridging the science-policy gap" it is definitely a valuable contribution for this journal. It is in the scope of NHESS and the SI as the paper focuses on drought impact assessment. However, as the readership of NHESS (compared to HESS) is certainly less familiar with hydrological modelling approaches and differences in groundwater model structures, more explanation on model setup, role of model**

We agree with Reviewer 1 that the study could also fit in, for example, HESS. In preparation of the submission, we carefully considered the options and found that the science and message of the paper fits best with this special issue. We are think that the additional paragraphs in the introduction and rephrased title will meet the wider readership of the NHESS.

**R1C9: Around the half of all Figures are placed in the Appendix. This is a nice approach to have additional information for the reader. However, reading the manuscript a lot of references were made to Fig. in the appendix even for major outcomes of the results and discussion section.**

This is true and perhaps inherent to a (virtual) modelling study including multiple scenarios and parameter testing in the sensitivity analysis. We have included more information regarding A1 in L211-217 (see also R2C2). In the revised manuscript, none of the main findings are based on figures in the appendix. References to appendix figures refer to specific conditions or alternative storage-discharge parameters. In the sensitivity analysis, we have revised the paragraph that refers primarily to Figure 6 (L358-364) with exception of sensitivity tested applying combined policy scenarios (L365-374 and L385-393). These results are relevant for the sensitivity analysis, but not important enough to be included in the main paper.

> *L358-364: Sensitivity tests show that the absolute groundwater storage in the large groundwater storage system is highly sensitive compared to the small groundwater storage system (time series shown in A5). However, this sensitivity has limited consequences for hydrological droughts in the large groundwater system, as drought duration and intensity increase slightly for each drought event (Figure 6). In the small groundwater system, hydrological drought duration nearly doubles when modelling longer response times (smaller storage-outflow parameters). Maximum hydrological drought duration increase from 137 days (baseflow) and 237 days (groundwater), to 273 and 455 days, respectively. These droughts also increase slightly in intensity, but much less compared to the drought duration (Figure 6).*

**R1C10: L232-241: Is the also a presentation of results according to the different storages models (Eq.4-6)? Different models will have more or less ability to buffer mild droughts. L284-L303: Again, more elaboration is needed here on the effect of the different groundwater models on the found drought characteristics. Is this included in the discussion section? L324-325 also suggests different effects of the same scenario for different groundwater systems (high, low etc).**

We thank the Reviewer for noting this. In short, yes, the different drought storage models result in a different decline of storage given the recharge and groundwater abstractions. This results in different hydrological droughts and overall groundwater storage availability. We have revised lines L284-303 (now L268-273) to emphasise the difference in decline.

> *L268-273: Groundwater storage in the large storage system shows a slower decline and therefore buffers more mild meteorological droughts compared to the other two systems, for which groundwater storage declines rapidly in summer months resulting in lower baseflow and ecological flow requirements in these systems. These results are similar for alternative storage-discharge parameters (A5), suggesting the difference is inherent to the different model structures.*

The revised sensitivity analysis (see also R1C9), we emphasise that the sensitivity of these systems is different regarding the absolute storage and the impact on hydrological droughts (L358-364). In some scenarios (combined 1-2-4), similar drought reductions are achieved, as shown in Figure A6. However, the impact of the policy is different, as the drought duration and intensity reduces slightly compared to the baseline in the small groundwater system, compared to a large reduction for the same scenario (compared to baseline) for all storage-outflow parameters in the large groundwater system (L365-374).

**R1C11: As there are major differences between high and low groundwater systems (suggested by Fig. 5) a comparison of both systems across the different drought metrics might be helpful.**

We agree with Reviewer 1 and we have presented drought characteristics in Table 4 for both systems. This table summarises mean hydrological drought duration, maximum intensity and frequency for baseflow and groundwater storage. In Figure 5 all drought events are graphically presented (duration in length of bar and intensity is coloured), whilst here only the mean of the baseline and combined scenarios are shown.

**R1C12: Water import is identified as important component in a future water management. Would be nice to have more discussion on potential ways to store redundant water (e.g., during winter high flows) to increase summerly low flows.**

This is an interesting point that we hadn't considered at first. When comparing the excess surface water storage (Qout) to the total surface water demand, there is only a small percentage of redundant water in the large groundwater system (2-3% of surface water demand) for baseline and combined scenarios. In the small groundwater storage system, increasing the reservoir would actually make a difference, as 22% of water demand is released as Qout, when storage capacity is exceeded. In combined scenarios this reduces to 7 and 16% for combined 1-2-3 and combined 1-2-4 respectively. We have included these additional discussion points to lines L262-264 and L459-463 to highlight the potential for reservoir operations and additional groundwater storage.

*L262-264: Excess surface water storage (Qout) represents a small proportion of in the large and medium groundwater system (2% and 5%) compared to 22% in the small groundwater system, suggesting larger reservoir storage might avoid the low reservoir levels that occur during mild droughts in the baseline.*

*L459-463: Alternatively, catchment-specific modelling could investigate if storing more surface water during winter in, for example, a small groundwater system, would aid to meet higher surface water demand in summer (??) or as additional groundwater recharge (?).*

**Minor comments:**

**R1C13: First sentence in the abstract is rather long, I suggest to split into two.**

Good point. We rephrased it in the revised manuscript.

*L1-3: Managing water-human systems during water shortages or droughts is key to avoid overexploitation of groundwater in particular. Groundwater is a crucial water resource during droughts sustaining both environmental and anthropogenic water demand.*

**R1C14: L24: 'sustained' means 'prolonged' here?**

This is a good suggestion, but we didn't mean only 'prolonged' in this opening sentence. Groundwater can be used to prolong water demand, complementing a shortage in surface water, but groundwater sustains current (and additional) water demand longer during a drought, as deficits in groundwater storage occur later compared to soil moisture and surface water deficits (L25-29).

**R1C15: L27: what is meant with "is available longer"?**

From this and the previous comment, it is evident that this phrasing is not entirely clear. We have rephrased the sentences to emphasise that groundwater storage is longer available due to the natural delay in drought propagation and delayed reduced groundwater recharge (L25-29).

*L25-29: Due to the natural delay in groundwater recharge, it may take weeks, months, or even years before a precipitation deficit propagates through the hydrological cycle, reducing groundwater recharge (Tallaksen and Van Lanen, 2004; Van, 2006). This natural delay results in groundwater storage being longer available compared to surface water, resulting in sustaining and complementing water demand during meteorological droughts (Taylor et al., 2013; Cuthbert et al., 2019).*

315 **R1C16: I suggest do move the sentence "In this study,..." (L31-33) to the end of the paragraph or to integrate it in a better way at the end of the introduction (L63-71) as here the aims/objectives of the study could be summarized.**

Thanks for the suggestion. We have rephrased this sentence to emphasise the focus of the study (R1C2) and have clarified the research objective in the later sentences (also in R1C2).

320 **R1C17: More elaboration on the term "conjunctive use of water" (L55) might be helpful here.**

We understand that using the term conjunctive use can refer to many different systems of integrated use of water resources. In this specific sentence, we refer to studies focusing on different aspects of socio-hydrology. To clarify this sentence, we have added '(or integrated)' and we have provided more explanation on conjunctive use when discussion the modelled third scenario (L237-239).

325

**R1C18: "large impact on streamflow droughts" (L59). Have Jaeger et al. (2019) performed a comparison of all the above-mentioned drought policy components (L53-57)? If so, that should be clarified. If not, I suggest a rephrasing "reservoir regulations and timely interventions have a larger impact on streamflow droughts than X, Y, Z...".**

We have implemented the suggestion and rephrased L59-60 (now L55-57).

330 *L55-57: Jaeger et al. (2019) are the first to model a set of drought policy measures aiming to conserve water. However, drought policy measures, either separately or combined, were found to have less impact on streamflow droughts compared to timely reservoir regulations.*

**R1C19: What is the share of surface water/groundwater use for the 13 of 18 companies (L79)? Could be summarized from Table A1 here.**

335 This information is used to inform the model and therefore included in section 3.2 Data. We have added a short phrase to refer to this section in L79 (now L86-88).

**R1C20: Trigger levels are communicated as percentages (e.g., L200-202) or as kind of return periods (e.g., L198). Return periods could be understand as probability. I wonder why SPI has then to be used? Of course, SPI as a trigger**
340 **level can also be converted to a probability, but is there an additional value to communicate SPI (as proxy for drought severity) instead of the probability (or percentiles?). Are there references or experiences from stakeholders that justify the SPI as a more valuable metric in drought assessment? Compared to that the reservoir triggers are nor communicated as a deviation from the mean and that reduces the clarity in trigger level communication here.**

We thank Reviewer 1 for pointing out the inconsistency in these lines that might be due to the large variability in trigger level
345 reporting in the thirteen drought management plans. In the management plans, a mixture of return periods, SPI and (reservoir) storage percentages are used (or at least reported). Drought severity is reported using either SPI or return periods for precipitation, whereas percentiles and percentages are often reported for reservoir levels. We have adopted the SPI levels to indicate the drought severity in the same manner as in the drought plans. Because reservoir levels, streamflow and groundwater levels are not standardised, trigger levels are indicated in percentiles. We rephrased these lines
350 and provided an example in L225-230.

*L225-230: Modelled trigger levels were based on averaged reported levels for precipitation anomalies (in monthly SPI). This average excludes reported extremely low SPI values (-2.32) or long return periods (100-150 year) for initial drought stages. Trigger levels are applied to precipitation (in SPI) and converted to percentiles for streamflow and groundwater level time series, as is common for the drinking water companies. For example, the*
355 *first category of drought management strategies can be activated due to a anomaly in precipitation, surface water or groundwater falls below the trigger level corresponding to a 1 in 8.5 year drought event (SPI < -1.18).*

**R1C21: Code availability: I suggest to enable open access to the code with at least a simple example code file (and data set) to reproduce the major parts of the analysis.**

We agree with Reviewer 1 that open access would be desirable. This is an aspiration, as more work is still required to meet
360 open science & code sharing standards. Unfortunately, this is difficult to meet within the time frame of current employment

contract. However, the code is available upon request, as indicated in the old (and revised) version of the manuscript.

**R1C22: L210: 80th percentage? Is this 80th percentile (exceedance level)? Is this percentile calculated on monthly basis or for the entire series?**

365 Correct, this should be percentile and is corrected in the revised manuscript. The percentile is calculated on a monthly basis, also now inserted in the text.

**R1C23: L219-L223: Would be helpful to have periods of reduced recharge etc. and the model spin-up also as color-coded information in Fig A2.**

370 That is a good idea. We have now changed Figure A2 to include the minor, moderate and severe meteorological droughts (see revised manuscript for new figure).

**R1C24: "the overall hydrological drought intensity and duration reduce for most scenarios" (L327) – here is missing something.**

375 Thanks for noting this incomplete sentence. We have now rephrased it in L365-367.

*L365-367: When running the drought management scenarios (combined scenarios only) with these different groundwater storage-outflow parameters, a reduction in the overall hydrological drought intensity and duration is evident for most scenarios (see Figure A6).*

**R1C25: What are examples of high costs for providers and users (L421) ?**

380 This sentence refers to either public/private drinking water services or water users that would need to investigate how to invest efficiently to reduce water demands permanently, as is highlighted by the cited papers. We have added 'drinking water providers and/or water users' to the sentence to clarify the emphasis.

**R1C26: "larger inter-annual storage" (L449), here is missing something or it should be a comparison with larger than?**

385 Correct, we added 'than the small groundwater storage system' to complete the sentence. Many thanks for this careful reading until the very end of the manuscript.

**Figures/Tables**

390 **R1C27: Many Figures have a nice formatting and I especially like the different point shapes (e.g. Fig 6). However, please remove the grey background in the Fig. to increase the readability (e.g. Fig A6).** We thank Reviewer 1 for this suggestion and have changed the grey background for a lighter outline to optimise readability of both the graph and light colour scheme of the point shapes in Figure 6 and A6 and lines in Figure A5.

395 **R1C28: Table A1: Please add numbers (1-13) to Table A1 to increase the connection to the numbers in the columns 3+6 in Table 1. I do not understand what "headroom" means here.**

We acknowledge that this has been confusing, as columns 3+6 in Table 1 do not refer to specific drought policies, but to the number of the drought strategies implemented. For example, out of the 13 drinking water companies, 6 companies apply water metering, but all of them (13) promote water use efficiency. We have revised the headers in Table 1 to clarify this point.

400 Regarding the use of the term 'headroom', we added a short description in the data section to defining headroom, as a proportion of water use depending on the long-term annual availability of water and referred to this calculation in equation 4 (see also R2C2). This is indicated by drinking water companies in the Data section and their proportional use of available water is reflected in the model (L221-225).

*L221-225: This water allocation percentage is also called 'headroom' by drinking water companies, as it indi-*
405 *cates remaining room given the long-term water availability and allocated water use. Between the drinking water*

*companies, water allocation varied between 82% and 95% (Table A1), resulting in an average of 88.5% used in the main result section to define the total anthropogenic water demand (Equation 4).*

**R1C29: Table 1: Why are some drought plan numbers (#) like 2 or 11 not mentioned in this table? Add 'yr' to the numbers in square brackets (e.g., average 8.5 yr, range 5 yr - 20 yr) for clarification or add return period in this column description.**

We have added 'yr' to the range in brackets in Table 1. The other suggestion refers to R1C28 and we assume that by revising the headers in Table 1 in R1C28, it becomes clear that the (#) refers to the number of companies implementing a certain drought management strategy.

**R1C30: Fig. 6: Try out a 2x2 panel instead of 4x1 and consider to add a regression line to evaluate the deviation of specific points from the "average". At the moment this Fig. is to wide compared to its height. Same is partly an issue in Fig A7.**

We thank Reviewer 1 for this suggestion and have changed Figure 6 accordingly. However, given the limited number of data points (drought events) for each parameter, adding a regression would not strengthen any conclusions in the manuscript.

**R1C31; Fig. 3: Stacked barcharts are critical here. Please move to a dodged version with 4 single bars for each category on the y-axis. Baseline, Scenarios and Combination could be placed in facets (4 facets in one column) to highlight the different groups here. 0%-label is missing and I cannot find the explanation for the dotted vertical line. The colored categories could also be placed into facets (if the dodged version doesn't work out).**

We thank Reviewer 1 for their suggestions. We have added the 0% label to the graph for consistency, but found that the stacked barcharts easier to interpret given they represent different proportions of the total water demand (in percentage). In the baseline, the proportions add up to 100%, which might not be met or exceed in different scenarios. This is one of the main messages of Figure 3. Another important message of Figure 3 is that the proportion of imported surface water is increasing in all scenarios that consequently results in lower (imported) groundwater use.

Please note that the dotted line represents the fixed percentage (6.9%) water import, as stated in the caption (third sentence). If additional water import is required, this dotted line is exceeded that is the case in all scenarios.

**R1C32: FigA6: Is it 12000 days in panel 2? Removing the outlier would improve the data representation (outlier comment could be added in the caption). Drought duration could be transformed to months (easier to read, but this is just a recommendation).**

We thank Reviewer 1 for their suggestion and have revised the figure and caption accordingly.

**Reviewer 2**

**I have found this manuscript to be well written, well structured and well presented. The topic is timely and I am very glad to see joining up of groundwater and water resources modelling, it happens far less than it should! Although the case study model is synthetic, a great deal of thought has gone into the both the model setup and parameter choices; the authors clearly have a strong working understanding of how UK water resource systems operate - and their synthetic model well reflects that. With the synthetic nature of the case study in mind, the authors are careful in their discussion and wider recommendations, making only recommendations that their sensitivity analysis supports. I have only some minor recommendations around improving clarity (primarily around presentation of the demand model) and some wider thoughts that may be addressed before the paper is published.**

We thank Reviewer 2 for these positive views and constructive comments to improve the submitted manuscript. We have revised relevant sections according to the suggested comments, please find the point-wise changes below in response to the minor comments.

**Minor comments**

**R2C1: As I mention, modelling water resources with groundwater in a joined up way is surprisingly rare - the authors may wish to include a slightly expanded review (even just a paragraph in the intro) on this subject to help justify their reasonably simplistic hydrological representation.**
We thank Reviewer 2 for valuing our integrated drought policy modelling and we have added a short paragraph as suggested (L59-69, see also R1C2). In this additional paragraph emphasised the need for including both surface water and groundwater, the additional complexity of groundwater droughts in natural and human-modified systems in response to R1C2 and R1C3.

**R2C2: Can water demand be written as an equation, just to make it easier to follow.**
This is a good point raised by Reviewer 2, as including an equation for the anthropogenic water demand is also more consistent with the overall modelling approach and improves the manuscript. We have inserted Equation 4 to clarify the definition of the total anthropogenic water demand. This also touches on the term 'headroom' as highlighted by Reviewer 1 (see R1C32) that is represented by the fraction $f_{dem}$ in Equation 4.

> *L154-158: The average annual runoff and groundwater recharge generated by the soil moisture balance also defines the total available water for anthropogenic water demand (ADem in mm $d^{-1}$)), following the water resource management plans in the case study area. Allocated ADem is defined as a fraction ($f_{dem}$) of the long-term average of annual runoff and groundwater recharge that is divided equally over the days of the year (Equation 4). $f_{dem}$ is defined by the proportional water use as reported by drinking water companies, see section 3.3 and Table A1 for more details.*

$$ADem = \frac{f_{dem} * (\overline{\sum Qr} + \overline{\sum Rch})}{365} \tag{1}$$

**R2C3: L116/133 - Perhaps either move some of the text from S3.2 here, or at least reference that this is described in more detail in the Data section. On first reading it appeared that this line was essentially the only description of water demand in the model!**
We agree with Reviewer 2 that there was some information missing in the submitted manuscript. We have improved the text by rephrasing the Model Structure section, also to include a 'Virtual socio-hydrological model' as suggested by Reviewer 1. In this new paragraph, both anthropogenic and environmental water demand are described (L125-139; see R1C5 for new paragraph). When introducing the model components, we added some text (L154-158) and introduced the demand equation 4 (see R2C2). These lines link the new paragraph with the Data section, elaborating on the averaged data of drinking water companies and additional information in the Supplementary information.

**R2C4: I was under the impression that ecological flows are typically met by reservoir releases, rather than groundwater pumping (though I suppose this is highly regional). No need to rerun the model, but might be interesting for UK readers.**
We thank Reviewer 2 for this interesting point that we had not included in the submitted manuscript. It depends on the catchment characteristics how ecological flows are met that can be by reservoir releases, as Reviewer 2 points out, but also by restricting groundwater abstractions supporting surface water and wetlands depending on the connectivity of the stream and aquifer (Environment Agency, 2019). Another approach is to pump groundwater from a deeper groundwater aquifer to meet river flows, also called augmentation scheme and applied in the UK (Shepley et al., 2009). Based on the thirteen drought management plans that source water from both surface water and groundwater, we found that groundwater abstractions are restricted when baseflow falls below a certain ecological minimum flow threshold. During severe droughts, drinking water companies can apply for a drought order to sustain groundwater abstractions potentially lowering ecological flows. This scenario is the reason for the modelled 'hands off flow' scenario applying this restricted use of groundwater. We have rephrased L239-243 in the Data section to emphasise the relevance of this scenario.

> *L239-243: The fourth scenario meets ecological flow requirements that aims to maintain baseflow in connected streams by reducing groundwater abstractions (also known as 'hands off flow': Environment Agency 2019). This*

example of maintaining ecological flow requirements is relevant to drinking water companies using both surface water and groundwater that might apply for drought permits reducing ecological flows during severe droughts (Environment Agency, 2016).

Reviewer 2 absolutely right that when simulating a specific catchment setting or alternative modelling approach, the 'hands off flow' also could include a fixed reservoir release or a combination of restricted groundwater use and reservoir release depending on the relevant catchment setting. We have mentioned these options in the Model limitation section (L484-486).

L484-486: The latter scenarios represents only restricting groundwater abstractions to meet environmental flow requirements that could be extended to a combination of reservoir releases and groundwater restrictions depending on relevant catchment characteristics (Environment Agency, 2019).

**R2C5: The groundwater and reservoir levels in this model are often 0 (Fig 2). This is no problem since the case study is synthetic, but the authors should note in the text that UK groundwater/reservoir systems are not this stressed (even if effort has been taken to parameterise the model in a sensible and nationally reflective manner) - perhaps expand a little in Section 5.3 (water companies might be alarmed if you give the impression that this is portrayed as a 'nationally average' model).**
We agree with Reviewer 2 that particularly the low groundwater storage system is quite stressed in the baseline scenario given the synthetic model settings. This is likely a combination of the small groundwater availability, fast responding aquifer that is modelled in a lumped model and considerable pressure on the water system *without any management interventions* in place. This is, as mentioned by Reviewer 2, not directly representative for the current status of water resources management in England. The first and primary reason for this difference is due to the baseline scenario setup that does not include any management interventions and has a fixed set of water use regardless of the actual water availability. For example, in the baseline scenario surface water, groundwater and imported surface water are used in a fixed proportion in the three groundwater systems (SW: 44.6%, GW: 48.5% and Qimp: 6.9%; described in L219-221). In reality water resource management is integrated, as ecological minimum flows are sustained applying a low flow alleviation scheme (Environment Agency, 2016; Howarth, 2018) and in larger water management regions (fixed) licences for surface water and groundwater can be used interchangeably (Thorne et al., 2003; Fowler et al., 2007; Shepley et al., 2009). Therefore, modelled groundwater and reservoir levels in scenarios that include either conjunctive use or 'hands off flow' are likely to be a closer representation of the status of water resources in England. As suggested, we have expanded on these aspects in section 5.3 (L470-475).

L470-475: When determining water availability for specific regions in England, the model runs should be revised using less generic, locally-relevant climate data. Moreover, given the range in local water resource availability and drought management practices (Table 1 and A1), current generic water resource management settings in the baseline might not represent all local water management strategies. Water resource availability in this model is based on annual available surface water and groundwater, implying that actual surface water storage and groundwater storage might be larger than shown here.

Another reason for the zero storage in the synthetic model is due to the lumped model setup. The lumped model approach implies that there is no groundwater flow represented outside the surface water catchment boundary. The groundwater module represents an isolated aquifer that is not connected beyond the surface water catchment boundaries nor has deeper aquifer layers to draw water from. These assumptions might results in a sharper decline in groundwater storage compared to reality, as pressure difference would introduce lateral flow, extending groundwater inflow beyond the surface water catchment boundaries or draw water from deeper aquifer layers when available. There is, however, modelled groundwater import (GSimp) component included in the model setup, although this is now only activated when the groundwater is depleted whereas in reality a difference in pressure would drive groundwater refill. In the manuscript these modelling assumptions are included in L202-205, but we have added these limitations to section 5.3 in L486-488 for consistency and to avoid misinterpretation of the results.

L486-488: A spatially-distributed model setup would also improve the representation of groundwater storage, as lateral groundwater flow is excluded in the lumped model setup. Inflow from deeper aquifer layers is limited to the imported groundwater component in the model.

**R2C6: Can the model/modelling setup be made publicly available so that results can be reproduced - it seems that the models/data are all openly available so I don't see why not?**
We agree with the Reviewer that this possible given the open data and use of published models. However, more work is still required to meet open science and coding standards, which are difficult to meet withing the time frame of the current employment contract.

**R2C7: Figure A2. soil moisture: can the green, light blue and dark blue horizontal lines be described in the caption.**
This is a good point raised by Reviewer 2, as currently information is missing in the legend and/or caption. We have added this information in the caption of Figure A2 (see track changes version of the manuscript). The green, light blue and dark blue lines represent the field capacity, critical moisture content and wilting point of the soil column as modelled by Van Lanen et al. (2013).

**R2C8: L204/Table 3: The increase in surface/ground water demand is actually an increase in water availability no? If I understand the reference to table 1, then it seems the licences are being maximised. Increasing demand as a response to drought seems confusing phrasing to me (why would a water company do that!). Maybe say +X% surface/ground water abstraction capacity? I would probably also describe this in a bit more detail in the text because it seems not trivial to conceptualise!** It is correctly interpret that the first scenario refers to existing licences being maximised or exceeded. Without any form of water demand management, water demand increases during droughts and water supply needs to increase to meet the rising water demand. Technically speaking, there is no difference between water use and water demand in the model, as both are included in water abstraction from the surface water or groundwater storage (see Figure 1). However, we value this suggestion and think that it would improve the clarity of the scenarios and Table 3 to increased water supply instead of water demand.
Given that we don't have access to actual licences or the usage of licences, we have modelled the indicated increase in water supply as an increase in water use percentage in addition to the current water use, as indicated in Table 1 (column $7^{th}$ first line). We did, however, include this limitation in the Discussion section in the submitted manuscript (now L489-491) stating that it would be advancing current results to apply dynamic water use that was not possible due to data limitations.

**R2C9: L207 - I guess conjunctive use is in contrast to a fixed proportion of demand met by surface water : groundwater? If so, I would state that here to make it clear.**
This is correctly assumed, as part of the reasoning behind the conjunctive use scenario is a flexible use of either surface water or groundwater storage. However, the application of conjunctive use of water resource was specifically mentioned in drought management plans (see Table 1) for six (out of thirteen) drinking water companies. Given the highlighted potential in academic literature and the wide application of conjunctive use, we decided to include a scenario in the model. As mentioned in L208 (now 237-238), conjunctive use is interpret as a flexible use of surface water and groundwater depending on the relative storage availability.

**R2C10: Figure 4: There is quite a lot to unpack here! I would rename the panels slightly to help the reader navigate the figure a bit more quickly:**
**-High groundwater storage system groundwater level (GWL)**
**-High groundwater storage system baseline GWL minus scenario GWL**
**-Low groundwater storage system GWL**
**-Low groundwater storage system baseline GWL minus scenario GWL**
We thank Reviewer 2 for this constructive comment and have changed the panel names as suggested.

**R2C11: L368: transferred, not traded!**
Thanks for spotting this mistake. We have corrected the mistake in the revised manuscript (L368 is now 408).

**R2C12: Finally, and feel free to ignore this as I find terminology to mainly be a subjective choice, the use of 'socio-hydrological model' in this context has left me a little confused. In terms of how sophisticated a representation of**

**hydrological process and sociological processes -this model is very asymmetric, with a minimal social representation. The feedback between hydrological and social is only the pre-defined water company drought response to lowering groundwater/reservoir levels (which I wouldn't class as a social process). By this definition, any water resources modelling application that includes a feedback between physical state variables and water consumption (and this is surely many/most -at least from my experience in water supply modelling) may be classified as sociohydrological (and if this is the case, then socio-hydrology is surely not an emerging field, as stated in L52!).**

We appreciate this comment and agree that the balance is slightly off between the hydrological model and water demand model. This study focuses on management impact on hydrological droughts that requires a rigorous representation in the lumped hydrological model setup, given the variability in natural hydrological droughts and possible varying impact of management influences (L62-69). However, given that this is one of the first studies applying drought policies to both surface water and groundwater use (L50-58), we think that the term socio-hydrological modelling is acceptable. Other listed modelled socio-hydrological studies in L50-58 show a wide range in their approach to represent management interactions or social response to water shortages and droughts. Given this context, we think that this study fits within that range, as we approached the sociological interaction with water shortages and drought in an indirect manner, namely by modelling water management strategies in response to droughts. We have revised the text slightly to remove the 'emerging field' statement related to socio-hydrology and focused on the modelling application of a specific measure of a drought policy instead (L50).

**References**

[revised manuscript text omitted]